# The effectiveness of a web-based Dutch parenting program to prevent overweight in children 9–13 years of age: Results of a two-armed cluster randomized controlled trial

**Emilie L. M. Ruiter**[1]*, **Gerard R. M. Molleman**[1], **Marloes Kleinjan**[2], **Jannis T. Kraiss**[3], **Peter M. ten Klooster**[3], **Koos van der Velden**[1], **Rutger C. M. E. Engels**[4], **Gerdine A. J. Fransen**[1]

1 Department of Primary and Community Care, Academic Collaborative Center AMPHI, Integrated Health Policy, ELG 117, Radboud University Medical Center, HB Nijmegen, The Netherlands, 2 Trimbos Institute, AS Utrecht, The Netherlands, 3 Department of Psychology, Health and Technology, University of Twente, AE Enschede, The Netherlands, 4 Department of Psychology, Erasmus University Rotterdam, DR Rotterdam, The Netherlands

* Emilie.Ruiter@Radboudumc.nl

**Data Availability Statement:** The dataset used in this manuscript is available from the Radboudumc

## Abstract

### Introduction

Although parental support is an important component in programs designed to prevent overweight in children, current programs pay remarkably little attention to the role of parenting. We therefore developed a web-based parenting program entitled "Making a healthy deal with your child". This e-learning program can be incorporated into existing overweight prevention programs. The aim of this study was to determine the effectiveness of this e-learning program.

### Materials and methods

The effectiveness was examined in a two-armed cluster randomized controlled trial. The participants were 475 parent-child dyads of children 9–13 years of age in the Netherlands who participated in an existing schoolclass-based overweight prevention program. At the school grade level, parents were randomly assigned to either the intervention or the control condition. Measurements were taken from both parents and children at baseline, and 5 and 12 months after baseline. Primary outcomes included the child's dietary and sedentary behavior, and level of physical activity. Secondary outcomes included general parenting style, specific parenting practices, and parental self-efficacy. Linear mixed effects models and generalized linear mixed effects models were conducted in R.

### Results

Intention-to-treat analyses and completers only revealed no significant effects between the intervention and control condition on energy balance-related behaviors of the child and

DANSEASY data repository at https://doi.org/10.17026/dans-22t-bgfz.

**Funding:** This study was funded by a grant from the Netherlands Organization for Health Research and Development (ZonMw; project number 505010296015, 200100001). This funder had no role in study design, data collection and analysis, interpretation of the data, decision to publish the results, or preparation of the manuscript.

**Competing interests:** The authors have declared that no competing interests exist.

**Abbreviations:** SES, Socio-economic status; EBRBs, Energy balance-related behaviors (EBRBs); FHB, Family-and-home-based; YHC, Youth Health Care; RCT, Randomized controlled trial; SSBs, Sugar-sweetened beverages; HES, Home Environment Survey; CONSORT, Consolidated Standards of Reporting Trials; IRR, Inter-rater reliability.

parenting skills after correction for multiple testing. The parents' mean satisfaction with the e-learning program (on a 10-point scale) was 7.0±1.1.

## Conclusions

Although parents were generally satisfied with the parenting program, following this program had no significant beneficial effects regarding the children's energy balance-related behaviors or the parenting skills compared to the control condition. This program may be more beneficial if used by high-risk groups (e.g. parents of children with unhealthy energy balance-related behaviors and/or with overweight) compared to the general population, warranting further study.

## Introduction

The growing number of children with overweight and obesity is a major health concern in industrialized countries [1–3]. In addition, in the Netherlands families with low socio-economic status (SES) and families of Turkish and Moroccan descent have a particularly high prevalence of children with overweight [4–8]. According to the Dutch Central Agency for Statistics, 16.4% of children 9–18 years of age were classified as overweight in 2020, a nearly three-fold increase in prevalence from 1980 in addition, 2.4% of children were considered obese [9]. The overall prevalence of overweight and obesity among Turkish and Moroccan children is 2–4 times higher than among Dutch children [7,10]. Moreover, preventing the development of overweight in children is important due to the high complexity of treating overweight [11], the increased likelihood of being overweight or obese in adulthood [12], and the detrimental health and social consequences associated with being overweight.

Parents play a key role in facilitating, supporting, and modeling their child's healthy—or unhealthy—energy balance-related behaviors (EBRBs), and in dealing with numerous environmental obesogenic factors [13,14]. In the long run, engaging in unhealthy EBRBs causes a chronic positive net energy balance in the child, which can result in the child becoming overweight. Reviews have increasingly emphasized the role of parenting in the development of childhood overweight and obesity [15–19]. Both a higher degree of parental involvement and helping parents encourage their children to engage in healthy EBRBs are considered effective techniques for preventing and/or treating childhood obesity and overweight [20–23]. In addition, to promote healthy EBRBs in interventions designed to prevent overweight in children, parenting styles and parenting practices are key components in achieving these changes in behavior and could make the interventions themselves more effective [15,16,19,21,24]. Thus, to ensure their children achieve a healthy weight, parents should apply *i*) adequate general parenting skills, and *ii*) specific parenting skills that encourage healthy EBRBs. Therefore, parents should be educated with respect to which EBRBs are healthy for their child [21]. In other words, an effective intervention should provide parents with information regarding *what* the children need (i.e., what is healthy for the child), as well as information regarding *how* to offer this as a parent [21]. General parenting includes the emotional climate in which a family functions [25], parents raise their child, and parents communicate with their child. Thus, support and control are the principal dimensions of general parenting skills [26,27]. Examples of specific parenting skills that encourage EBRBs can include monitoring their child's EBRBs, modeling healthy EBRBs, and setting rules for their child regarding EBRBs.

 

Despite the clear role of parenting, most Dutch and international obesity-prevention programs developed to date pay only limited attention to parenting aspects [21,28–33]. In addition, the few interventions that included parenting are family-and-home-based (FHB) interventions [34–36], which were offered primarily face-to-face in the form of either group meetings or one-on-one meetings with a healthcare professional. Thus, these FHB interventions are limited due to their time-intensive nature (and the resulting relatively low parental response) [37], as well as a reduced ability to provide the intervention at times that best suit the parents' needs.

Offering parents an online intervention can help increase parental participation [38]. Moreover, systematic reviews have shown that family-based eHealth interventions may be promising as part of a primary prevention-based strategy designed to improve children's health [39,40]. In addition, online programs have the potential to: *i*) engage high-risk parents; *ii*) maximize reach by overcoming barriers such as a limited availability of trained professionals, geographic restrictions, logistics, social stigma, and distrust; and *iii*) reduce delivery costs [41]. In the Netherlands, no existing intervention is currently available for parents to follow in their own home at a time that suits them best. To close this gap, we developed a fully self-guided web-based parenting program entitled "Making a healthy deal with your child", a brief, flexible program that is simple to use, appealing to parents from a variety of demographic backgrounds, and can be incorporated easily into existing interventions for preventing overweight in children.

The purpose of this e-learning program is to improve upon existing interventions by: *i*) strengthening both the general parenting style and specific parenting practices, and *ii*) increasing the self-efficacy of parents of children 9–13 years of age [42]. Specifically, this e-learning program is designed to teach parents how to encourage and support their child's decisions with respect to developing and maintaining healthy EBRBs, as well as how to handle situations in daily life that can jeopardize healthy EBRBs. The theory, content, and structure of this program were derived from the online course entitled "Talking with your child" [43] and are described in detail in our previously published study protocol [44]. In summary, the theory behind this e-learning program is based on the theoretical insights obtained from Parent Effectiveness Training and the Parent Management Training-Oregon Model [45,46]. Using a six-step model, our program shows parents how to improve their communication and problem-solving skills [44]. For the theoretical knowledge regarding dietary, sedentary, and physically active behaviors, we used the guidelines established by the Netherlands Nutrition Center [47] and the Dutch Norm for Healthy Physical Activity [48], which were also used by the Dutch Youth Health Care (YHC) system. The e-learning program's content was based on the results of focus groups conducted with mothers who live in a low-SES neighborhood in the Netherlands [49]. We chose parents who live in a low-SES neighborhood because these parents are historically difficult to reach with an intervention [50], and childhood overweight is more prevalent in these neighborhoods [5]. In addition, we identified the following five difficult everyday life situations related to healthy EBRBs: *i*) children having insufficient time in the morning to eat breakfast; *ii*) the daily struggle at the dining table (children not wanting to eat their vegetables); *iii*) the child's continuous desire to eat candy and unhealthy snacks; *iv*) children spending an excessive amount of time watching television; and *v*) children spending an excessive amount of time using the computer and not wanting to turn off their computer (e.g., when playing a computer game) when told to do so. We reasoned that if we use these situations and the underlying reasons why parents experience difficulties, many parents would recognize these situations and would be interested in learning parenting skills that would teach them how to deal with these difficult situations in daily life. Thus, we reasoned that many parents would feel more compelled to complete the e-learning program.

The aims of this study were to investigate the effects of this web-based parenting program on: *i*) dietary, sedentary, and physically active behaviors among children 9–13 years of age who were participating in the existing school-based overweight prevention program entitled "Scoring for Health"; and *ii*) the parenting styles, EBRB-specific parenting practices, and parental self-efficacy of the parents of these children.

Here, we tested the hypothesis that at 5 and 12 months, the children of parents who received the e-learning program would: *i*) have a healthier diet (e.g., they would eat more fruit and vegetables, eat breakfast more often, and drink fewer sugar-sweetened beverages); *ii*) be less sedentary (e.g., would engage in less screen-viewing time); and *iii*) have a higher level of physical activity compared to both their baseline values and the children whose parents did not receive the e-learning program (i.e., the control condition). The secondary objectives of the e-learning program include strengthening parenting styles, improving specific parenting practices (monitoring, modeling, and applying rules regarding EBRBs), and increasing parental self-efficacy.

## Materials and methods

### Design of the study

The effectiveness of this web-based parenting program was studied in a two-armed (intervention versus control) cluster randomized controlled trial (RCT).

The study was approved by the Medical Review Ethics Committee of the Arnhem-Nijmegen region in the Netherlands (registration number 2012495, NL4280309112) [44], and the trial is registered with the Dutch Trial Register (NTR3938). This trial is registered at the Dutch Trial Register (NTR3938). All authors confirm that all ongoing and related trials for this intervention are registered.

### Participants and protocol

Recruitment and enrolment of participants began at the end of 2012. The baseline data were collected from January 2013 through December 2013. After the baseline data were collected the participants were randomized in February 2013 and September 2013. For each parent–child dyad, the follow-up measurements were collected at fixed time points, 5 and 12 months after the baseline measurements. All parents and children received written information regarding the study [44]. In addition, one researcher (author ER) was present at all classes to provide verbal information regarding the study and answer any questions raised by the children. For this study, we used passive parental consent, which was approved by our Medical Review Ethics Committee.

**Inclusion criteria.** We recruited parent-child dyads in the Nijmegen region in the Netherlands. To be eligible, the children had to be 9–13 years of age, in the fourth, fifth, or sixth grade of regular primary school, and participating in the existing school-based overweight prevention program entitled "Scoring for Health". The parents and children were required to both speak and read Dutch. We selected children participating in the "Scoring for Health" program as an illustrative example, as this is a school-based intervention certified by the Center for Healthy Life (in Dutch, the *Centrum voor Gezond Leven*) in the Netherlands. Moreover, arrangements have been made to offer this intervention to a large group of children each year. Twice a year in the Nijmegen region, children 9–13 years of age are selected to participate in the "Scoring for Health" intervention program, yielding a total of 500 participating children a year.

The aim of the "Scoring for Health" program is to increase the awareness of primary school students (and their parents) regarding the importance of engaging in a healthy lifestyle. The program runs for 20 weeks and begins and ends with a sports clinic at a nearby professional soccer club.

**Recruitment.** Eleven primary schools in the Nijmegen region in the Netherlands participate in the intervention program "Scoring for Health", and the principals of these 11 schools were asked to participate in our study. Specifically, we asked the principals' permission to distribute information to the parents of children in the appropriate grades. The parent-child dyads who were assessed as eligible received a baseline questionnaire as reported previous [44].

**Randomization.** After the baseline measurements were collected, an independent researcher at the Behavioral Science Institute in the Netherlands randomly assigned all participating school classes to either the intervention condition or the control condition. Randomization was performed centrally at the school grade level class and within schools (to control for school characteristics) using a computerized random number generator with a blocked randomization scheme (block size was 2); the conditions were stratified by ethnicity. If more than one child living in the same household participated in the study, all participating children in that household were assigned to the same condition as the oldest participating child in order to avoid contamination between conditions.

**Sample size.** The sample size calculations for this study have been reported previously [44]. In brief, using a two-sided test with alpha = 0.05 and a power of 0.80, and taking into account the clustering of children within classes. We assume an average cluster size of 14 and an intraclass correlation coefficient (ICC) of 0.05, based on a study on ICCs for obesity indicators in primary schools in which the ICCs for most outcomes varied between 0.02 and 0.05 [51]. We calculated that we needed 161 parent-child dyads in each condition (i.e., 161 in the intervention condition and 161 in the control condition) in order to detect an increase of 20% in the number of children who meet Dutch standards with respect to dietary, sedentary, and/or physical activity behavior, given an initial compliance rate of 40% (i.e., an increase from 40% to 60%). The assumption of an increase of 20% is based on previous effectiveness studies and school-based programs in children 9–13 years of age [52–55].

**Program delivery.** We sent parents in the intervention condition a personal login code for the e-learning program by email, and parents received a standard brochure from the Dutch Nutrition Center regarding healthy eating and physical activity [56]; this brochure is also distributed by the Dutch YHC organizations. The parents in the control condition received only the brochure. Over a course of ten weeks, the parents in the intervention condition were allowed to complete the e-learning program at their own pace. After finishing each module, the parents received an e-mail to thank them for finishing the module and to encourage them to complete the next module. In addition, we sent reminder e-mails 7 and 12 weeks after providing the log-in code, reminding parents to complete the e-learning program.

**The intervention.** The e-learning program "Making a healthy deal with your child" consists of five 30-minute modules that are based on five difficult situations encountered in daily life. Each module consists of video fragments in which "good" and "less good" examples of communicating with a child are shown, followed by the six-step problem-solving model, practical and theoretical assignments, and feedback. During the program, parents receive tools that they can use to encourage their child to develop a healthy diet, to be less sedentary, and to engage in regular physical activity during everyday situations. Importantly, the parents can follow the program in their own home, at a time that suits them best.

## Measurements

Measurements were collected from the children and parents in both conditions at baseline and again 5 and 12 months after the baseline assessment, as reported previously [44]. The questionnaires were developed using existing validated Dutch questionnaires (or, if no validated

questionnaire was available, questionnaires that were used in currently ongoing projects within the Netherlands) [44].

**Socio-demographic and other characteristics of the children and parents.** The following socio-demographic data were obtained at baseline: the child's age, gender, and ethnicity, the participating parent's age and ethnicity, and the highest level of education achieved by both parents, providing an indicator of the family's SES [57]. Ethnic background was determined by asking the country of birth of the child and both parents. The parents were able to choose from the following six categories: "*the Netherlands*", "*Turkey*", "*Morocco*", "*Suriname*", "*Netherlands Antilles*", or "*other country*". The results were classified using standard procedures established by the Dutch Central Agency for Statistics [58]. The parents' level of education was based on the highest level achieved of either parent and was classified as follows in accordance with international classification systems [59]: "low" (lower general secondary education, lower vocational training, or primary school), "middle" (intermediate vocational training, higher general secondary training, or pre-university education), and "high" (higher vocational training or university education). The parents' perception of their child's weight status was determined at baseline by asking the following question: "*What do you think of* your child's *weight*?*"* The parents were instructed to select the answer that they felt best suited their child's weight status, and the answers were then classified as follows: "extremely low", "low", "normal", "heavy", or "extremely heavy".

**Energy balance-related behaviors (EBRBs).** The children's dietary behavior was assessed by asking both the children and parents how often they consume the following items: *i*) breakfast, *ii*) snacks, *iii*) vegetables, *iv*) fruit, and *v*) sugar-sweetened beverages (SSBs). Eight responses were possible, including "*never*", "*1 day a week*", "*2 days a week*", etc., and the amounts were measured in pieces (fruit), serving spoons (vegetables), and glasses (SSBs) per day.

The child's sedentary behavior was assessed by asking both the children and parents how many days per week the child: *i*) watches television/videos/DVDs, etc.; and *ii*) uses the computer (including accessing the Internet and playing computer games). Eight answers were possible, including "never", "*1 day a week*", "*2 days a week*", etc. In addition, the amount of time the child spends on these activities on an average day was measured, with the following possible answers: "*less than half an hour a day*", "*half an hour to 1 hour per day*", "*1–2 hours per day*", "*2–3 hours a day*", and "*3 or more hours a day*".

The child's physical activity was assessed by asking the both children and parents how many days per week the child: *i*) plays outside, *ii*) participates in a sport at a sports club, *iii*) has gym class at school, and *iv*) goes to school by walking or riding a bicycle. Eight answers were possible, including "*never*", "*1 day a week*", "*2 days a week*", etc. In addition, the amount of time the child spends on these activities on an average day was measured, with the following possible answers: "*less than half an hour a day*", "*half an hour to 1 hour per day*", "*1–2 hours per day*", "*2–3 hours a day*", and "*3 or more hours a day*".

All of the child's EBRBs were then dichotomized as "meets" or "does not meet" the Dutch standard for healthy EBRBs. According to Dutch standards [47,48,56], healthy EBRBs in children include: *i*) eating breakfast daily; *ii*) eating at least two pieces of fruit daily; *iii*) eating at least three serving spoons of vegetables daily; *iv*) drinking less than two glasses of SSBs daily; *v*) less than two hours of screen time (watching television and/or using the computer) each day; *vi*) playing outside for at least one hour each day; and *vii*) engaging in an organized sport at least twice a week for ≥30 minutes each.

**Parenting dimensions.** A full description of the scales, scale properties, items, reliability, and examples of items in the parenting dimensions in our sample is provided in Additional File 1 (S1 Appendix). The internal consistency of the instrument ranged from acceptable to

good, with Cronbach's alpha values ranging from 0.62 to 0.80 for the different scales [60], with the exceptions of the parental scale for sedentary behavior role modeling (2-item scale) and physical activity role modeling (6 items), which had Cronbach's alpha values of 0.41 (unacceptable) and 0.59 (poor), respectively. Total scores were calculated when at least 75% of the questions were answered. The general parenting style was measured using the validated Dutch translation of the Steinberg parenting style instrument [25,61]. This 15-item instrument assesses two parenting style dimensions: the parents' involvement and general strictness. The following four parenting styles have been established based on these two parenting dimensions: authoritative (high involvement, high strictness), permissive/indulgent (high involvement, low strictness), authoritarian (low involvement, high strictness), and neglectful (low involvement, low strictness). In addition, we generated these four parenting styles by dichotomizing the sample on each dimension (median split) and examining the two dimensions simultaneously.

**Parenting practices.** The Parental Feeding Style questionnaire [62,63] was completed by both the children and their parents. This questionnaire consists of 27 items used to measure four distinct parental feeding styles: instrumental feeding (i.e., using food to regulate the child's behavior), emotional feeding (i.e., providing food in response to emotional distress), encouragement to eat (encouraging food variety and interest in food), and control over eating (parental restrictions). Parental policies regarding physical activity (monitoring the child's physical activity behavior) were assessed via the parental questionnaire, with five items derived from the Dutch translation of the validated Home Environment Survey (HES) [64]. Parental modeling of dietary, sedentary, and physical activity behavior) was assessed via the parental questionnaire, with 20 items derived from the Dutch translation of the HES [64]. Parental EBRB rules were assessed via the parental and child questionnaire. Seven items were derived from the Child Monitor Questionnaire, which is part of the "Local and National Health Monitor" in the Netherlands [57]. Children and parents were asked whether the parents had rules and/or agreements with their child regarding the following eight aspects: *i*) eating breakfast each morning, *ii*) consuming snacks, *iii*) eating vegetables, *iv*) eating fruit, *v*) drinking SSBs, *vi*) total hours per day spent watching television, DVDs, or other videos, *vii*) total hours per day spent using the computer (including accessing the Internet and/or playing computer games), and *viii*) total hours per day spent playing outside. The children and parents were instructed to answer each question with the following possible answers: or *i*) yes, and we stick to them ("*strict rules*"); *ii*) yes, but we are flexible with them ("*indulgent rules*"), or *iii*) no, we have no rules about it ("*no rules*"). Parental self-efficacy was assessed in the parental questionnaire, with 16 items derived from the Parenting Sense of Competence [65], and was used to measure two dimensions, namely efficacy and satisfaction.

**Process evaluation.** We monitored the parents' willingness to follow the e-learning program and the parents' satisfaction with the e-learning program. The parents' login activity in the e-learning program was monitored during the intervention period. This allowed us to monitor which parents started the e-learning program (and which parents did not). This approach also provided information regarding the number of modules each parent completed (dose received) of the in total 5 modules delivered in the e-learning program (dose delivery). Parental satisfaction with the intervention was assessed in the questionnaire given at 5 months by asking "How would you rate the intervention on a scale from 0–10?"

**Anthropometry.** The children's height and weight were measured at baseline (i.e., at the start of the "Scoring for Health" program) using a calibrated scale and measuring tape in accordance with established guidelines [66]. All measurements were performed by a YHC professional or trainee, as reported previously [44].

## Statistical analysis

We used the Chi-square test and the independent Student's *t*-test in IBM SPSS Statistics 25 [67] to determine whether randomization resulted in a balanced distribution of demographic variables, weight-related characteristics, and baseline primary and/or secondary outcomes. Continuous variables are presented as the mean, and categorical data are presented as the percentage of respondents within each of the possible categories. The degree of agreement among the answers given by the children and parents—the inter-rater reliability—was calculated using the intraclass correlation coefficient (for continuous variables) or Cohen's kappa (for categorical data). The value for inter-rater reliability ranges from 0 to 1 as follows: 0, no agreement; <0.2, none to slight agreement; 0.2 to <0.4, fair agreement; 0.4 to <0.6, moderate agreement; 0.6 to <0.8, substantial agreement; 0.8 to <1, almost perfect agreement; and 1, full agreement [68]. Moreover, the variables for which the distributions differed between the two conditions were entered as covariates in all models performed to test the effectiveness of the e-learning program. Further, we used the Chi-square test and the independent Student's *t*-test as well to determine whether there was a selective loss to follow-up regarding demographic variables, the weight status of the child and/or parent, and the children's EBRBs and the parenting dimensions. The effects of the e-learning program on changes in the child's dietary behavior, sedentary behavior, and physical activity, as well as the parenting style, parenting practices, and parental self-efficacy, were tested in accordance with the intention-to-treat principle (all participants are analyzed in the condition to which they were randomly assigned). Sensitivity analyses were performed using completers-only data (i.e., only the parent-child dyads who completed all three questionnaires). To determine the effectiveness of the intervention, we used linear mixed models in R [69], instead of the planned regression analyses in Mplus as described in the study protocol [44]. Linear mixed models were chosen for two reasons: *i)* because of the nested structure of the data (i.e., repeated measurements nested within participants, and participants nested within schools), and *ii)* because multi-level analyses do not delete participants listwise and can more appropriately handle data missing at random [70,71]. The multi-category variables for parental EBRB rules were dichotomized in order to compare between "no rules" and "indulgent/strict rules" and between "strict rules" and "no/indulgent rules", providing better interpretation of the data. Linear mixed models for continuous outcomes and generalized linear mixed models with logit links for binary outcomes were used. The package lme4 was used for both LMMs and GLMMs [72]. All models included a random effect for students to account for repeated measurements within participants. For each outcome, it was also tested whether it is necessary to additionally account for nesting of students within schools. This additional random factor was only included if the intraclass correlation for the school-level was higher than .05, suggesting significant clustering of students within schools [73,74]. In addition, it was tested whether a random intercept and slope model significantly better fit the data than the more parsimonious random intercept model using log-likelihood ratio tests. If this was the case, the model with random slopes was used. If the more complex models including additional random factors and/or random slopes did not converge or were singular, we used a more parsimonious model, for example by only modelling random intercepts. Time, condition and their higher-order interactions were included as fixed effects in all models. An unstructured covariance structure was used for all models. Because there were two possible starting points of the study, namely *i)* February 2013, in which participants of 9 schools were randomized, and *ii)* September 2013, in which participants of another 2 schools were randomized, the variable 'moment of randomization' was included as covariate to correct for possible seasonal effects. To adjust for multiple testing, the p-values of the time x condition interaction effects were adjusted using Holm-Bonferroni sequential correction. All

models were visually inspected for homogeneity and normal distribution of errors using residuals versus fitted value plots and Q-Q plots. If it was suspected that these assumptions might have been violated, or if a continuous outcome was strongly left or right skewed, we additionally fitted the models with beta-regression using the glmmTMB package [75]. The results from the beta-regression models were then compared with linear mixed models. Since the outcomes from the linear and beta models did not lead to substantially different conclusions, only linear mixed models results were reported for continuous outcomes. To determine treatment effects at each of the follow-up timepoints, pairwise post-hoc comparisons of estimated marginal means by time and condition were performed using the emmeans package. For pairwise comparisons, p-values were adjusted using the Tukey-method to account for multiple testing. An α of 0.05 was used to indicate statistical significance for all analyses. The results have been reported in accordance with CONSORT (Consolidated Standards of Reporting Trials) guidelines [76,77].

## Results

### Study participation

Of the 548 parent-child dyads who were assessed for eligibility, 475 (87%) completed the baseline questionnaire (T0), 360 of whom (76%) participated in either the intervention or control condition and completed the 5-month follow-up assessment (T1). Finally, 341 of the 475 participants (72%) completed the 12-month follow-up assessment (T2); see Fig 1.

In the entire study population, we found no selective loss to follow-up with respect to the parent or child's age, gender, ethnicity, or weight status, or the parent's level of education. When we examined loss to follow-up in the two conditions, we found that the loss to follow-up was higher in the intervention condition than in the control condition ($p = 0.001$). We found no selective loss to follow-up in the intervention condition. In the control condition, however, we found that the loss to follow-up was higher among non-Caucasian children ($p = 0.010$ vs. Caucasian children) and children that were overweight ($p = 0.024$ vs. children that were not overweight).

### Demographics and weight-related characteristics

The baseline demographic characteristics of all 475 parent-child dyads are summarized in Table 1. We found no difference between the intervention and control conditions; moreover, we found no difference between conditions of the completers-only as well (see S2 Appendix). Overall, approximately 7% of the participating children were non-Caucasian, a third of the parents had a low level of education, one-fifth of the children were overweight or obese, and nearly half of the parents were overweight or obese.

### Children's EBRBs and parenting characteristics

The children's EBRBs and the parenting characteristics at baseline are summarized in Table 2. We found no statistically significant difference between the intervention and control condition, with the exceptions of the child's "SSB intake" according to the parents, and "sports". Moreover, we found no difference in "SSB intake" in the completers-only (see S3 Appendix); therefore, we excluded the variable "sports" from further analyses. According to both the children and the parents, more than 80% of the children eat breakfast daily; however, less than 30% of the children eat vegetables daily, eat 2 or more portions of fruit daily, drink fewer than 2 glasses of SSBs daily, and/or play outside at least 1 hour each day.

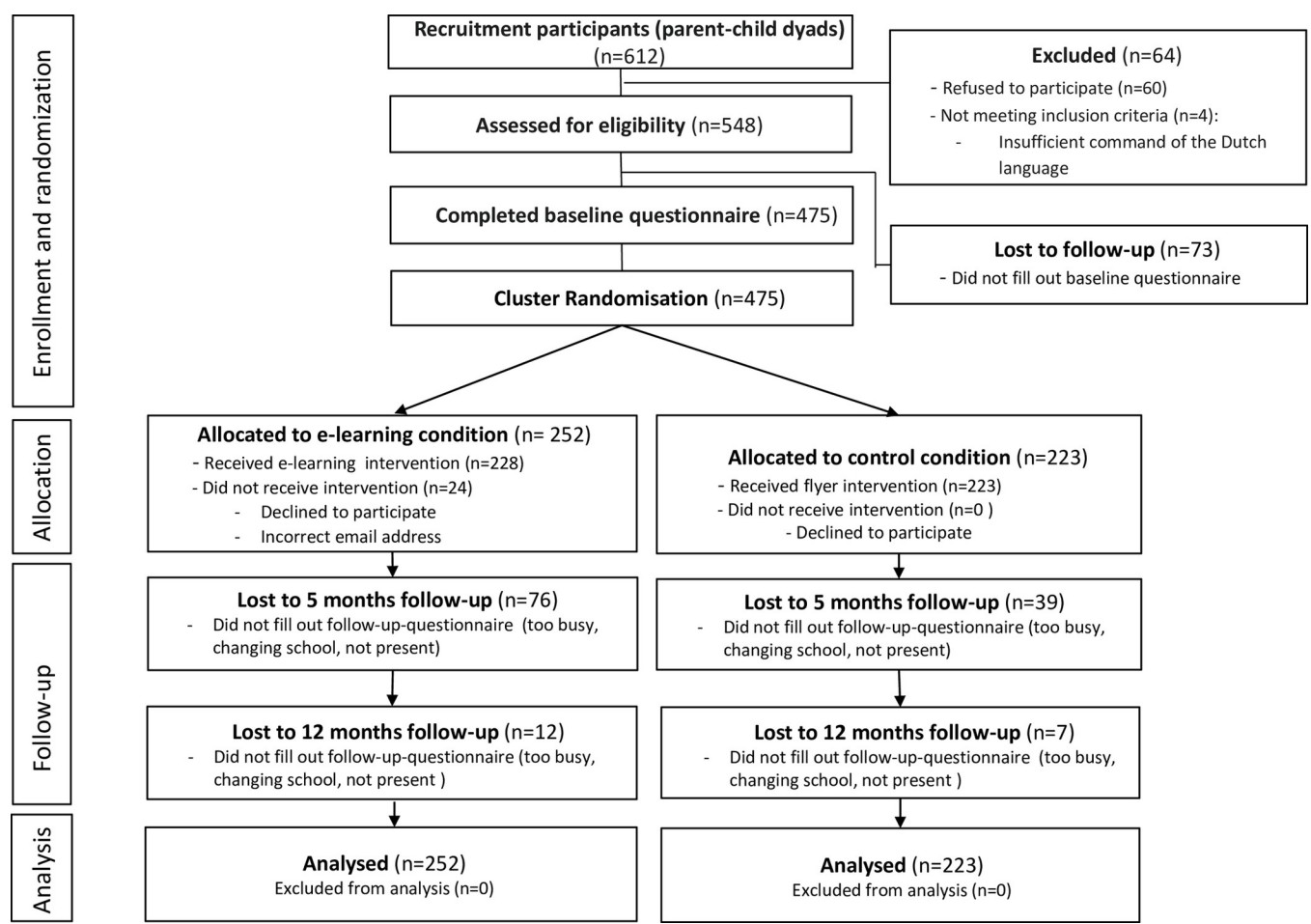

**Fig 1. CONSORT flow diagram depicting the enrolment, allocation, follow-up, and analysis for the indicated number of parent-child dyads**\*. \*In the completers-only framework, 164 and 177 parent-child dyads were analyzed in the intervention and control condition, respectively.

Interestingly, the children indicated that their EBRBs are healthier compared to what their parents indicated. In addition, 68% of the children reported that engage in ≤120 minutes of screen time per day, compared to only 36% of parents who reported that their child engages in ≤120 minutes of screen time per day. Overall, the degree of the inter-rater reliability (IRR) among the children and parents as reported in Table 2 ranged from no agreement to fair agreement, except for the variables vegetables eaten per week, minutes playing outside per week, and minutes of physical active per week, which had moderate agreement; moreover, IRR indicated substantial agreement with respect to sports. Finally, >80% of parents reported that they have strict rules regarding their child eating breakfast, whereas approximately 30% of parents reported that they have no rules regarding watching television and >60% of parents reported that they have no rules regarding playing outside.

## Effects of the e-learning program

The effects of the e-learning program on the children's EBRBs and parenting dimensions according to the results of the intention-to-treat analysis are summarized in Tables 3 and 4, and the completers-only sensitivity analyses are shown in Additional file 4 (S4 Appendix).

**Table 1. Demographics and weight-related characteristics of the entire study population.**

| | Intervention condition N = 252 N (%) | Control condition N = 223 N (%) | *p*-value |
|---|---|---|---|
| **Child** | | | |
| **Age, years (mean ± SD)** | 10.4 ± 1.1 | 10.4 ± 1.1 | 0.802[a] |
| **Gender** | | | |
| Male | 136 (54.0) | 117 (52.5) | |
| Female | 116 (46.0) | 106 (47.5) | 0.743[b] |
| **Ethnicity** | | | |
| Caucasian | 242 (96.1) | 208 (93.3) | |
| Non-Caucasian | 10 (3.9) | 15 (6.7) | 0.592[b] |
| **Weight status based on BMI** | | | |
| Not overweight | 205 (81.7) | 177 (79.4) | |
| Overweight | 38 (15.1) | 41 (18.4) | |
| Obese | 8 (3.2) | 5 (2.2) | 0.546[b] |
| **Parent** | | | |
| **Age, years (mean ± SD)** | 42.4 ± 4.8 | 42.3 ± 4.8 | 0.828[a] |
| **Ethnicity** | | | |
| Caucasian | 94.4% | 92.8% | |
| Non-Caucasian | 5.6% | 7.2% | 0.622[b] |
| **Education level** | | | |
| Low | 79 (31.5) | 72 (32.9) | |
| Middle | 127 (50.6) | 107 (48.9) | |
| High | 45 (17.9) | 40 (18.3) | 0.928[b] |
| **Weight status based on BMI** | | | |
| Not overweight | 129 (54.2) | 121 (55.8) | |
| Overweight | 75 (31.5) | 60 (27.6) | |
| Obese | 34 (14.3) | 36 (16.6) | 0.603[b] |

Unless indicated otherwise, data are presented as N (%).

[a] Student's *t*-test;

[b] Chi-square test.

Based on both the children's responses and the parents' responses to the questionnaires, and after adjusting for multiple testing, we found no significant differences in the changes over time between the intervention and control condition with respect to the children's diet, sedentary behavior or physical activity behavior. In addition, post-hoc pairwise comparisons showed no significant difference between conditions at any of the follow-up times.

### Effects of the e-learning program on parenting dimensions

Based on both the children's responses and the parents' responses to the questionnaires, and after adjusting for multiple testing, we found no significant differences in the change over time between the intervention and control condition with respect to parenting dimensions, including general parenting, specific EBRB-guided parenting (e.g., monitoring, modeling, and establishing EBRB rules), or parental self-efficacy. In addition, post hoc pairwise comparisons showed no significant difference between conditions at any of the follow-up times.

**Table 2. Baseline Characteristics of the children's EBRBs and the parenting dimensions and inter-rater reliability (IRR) within n = 475 parent-child dyads.**

| | According to the children | | According to the parents | | Agreement among answers parent-child |
|---|---|---|---|---|---|
| | Intervention N (%) | Control N (%) | Intervention N (%) | Control N (%) | IRR |
| **Dietary behavior (mean ± SD)** | | | | | |
| Days of having breakfast per week | 6.6 ± 1.1 | 6.6 ± 1.2 | 6.7 ± 0.8 | 6.7 ± 0.8 | 0.38 |
| Portions of fruit per week | 15.6 ± 9.3 | 14.9 ± 9.1 | 11.1 ± 6.8 | 10.8 ± 7.4 | 0.45 |
| Serving spoons of vegetables per week | 12.6 ± 6.1 | 12.7 ± 6.3 | 10.1 ± 4.6 | 10.2 ± 4.7 | 0.25 |
| Glasses of SSBs per week | 31.9 ± 25.4 | 28.8 ± 23.9 | 23.3 ± 14.6 | 22.4 ± 14.7 | 0.27 |
| **Dutch standards for dietary behavior** | | | | | |
| Daily breakfast | 209 (82.9) | 188 (84.3) | 216 (86.1) | 185 (84.1) | 0.31 |
| Daily vegetables | 70 (27.8) | 66 (29.6) | 59 (23.7) | 43 (19.5) | 0.21 |
| Daily 2 pieces of fruit | 51 (20.2) | 40 (17.9) | 14 (5.6) | 9 (4.1) | 0.20 |
| Daily <2 glasses of SSBs | 53 (21.3) | 55 (25.1) | 54 (22.9) | 64 (31.4)* | 0.31 |
| **Sedentary behavior and physical activity (mean ± SD)** | | | | | |
| Minutes screen time per week | 707.5 ± 615.8 | 777.7 ± 603.4 | 1003.9 ± 479.2 | 1023.7 ± 446.9 | 0.26 |
| Minutes playing outside per week | 600.1 ± 433.0 | 623.3 ± 462.6 | 479.8 ± 317.7 | 469.8 ± 318.2 | 0.44 |
| Minutes physical active per week | 1095.1 ± 616.2 | 1050.4 ± 579.5 | 941.5 ± 440.7 | 893.0 ± 394.4 | 0.44 |
| **Dutch standards for sedentary behavior and physical activity** | | | | | |
| Daily ≤120 minutes screen time | 171 (68.1) | 144 (65.2) | 92 (36.9) | 82 (37.3) | 0.19 |
| Dutch standard healthy Exercise | 224 (89.2) | 189 (84.8) | 223 (89.2) | 199 (90.5) | 0.29 |
| Play outside ≥1 hour daily | 73 (29.2) | 68 (30.8) | 38 (19.0) | 36 (18.7) | 0.23 |
| Playing sport 2 days a week | 199 (79.0) | 144 (64.9)^ | 205 (81.7) | 150 (67.9)^ | 0.70 |
| Playing sports 2 days a week ≥30 minutes | 197 (78.5) | 139 (62.9)^ | 199 (79.6) | 146 (66.4)^ | 0.66 |
| **General parenting** | | | | | |
| Authoritative parenting style | - | - | 82 (33.1) | 65 (29.4) | - |
| Authoritative parenting style (arbitrary) | - | - | 234 (94.4) | 204 (92.3) | - |
| **Monitoring (mean ± SD) Parental feeding style score** | | | | | |
| Control over eating | 3.6 ± 0.63 | 3.6 ± 0.68 | 4.1 ± 0.49 | 4.1 ± 0.42 | 0.22 |
| Emotional feeding | 1.6 ± 0.66 | 1.5 ± 0.69 | 1.3 ± 0.45 | 1.3 ± 0.44 | 0.15 |
| Encouragement to eat | 2.8 ± 0.81 | 2.8 ± 0.81 | 3.7 ± 0.57 | 3.6 ± 0.58 | 0.17 |
| Instrumental feeding | 1.7 ± 0.72 | 1.7 ± 0.70 | 1.5 ± 0.57 | 1.5 ± 0.57 | 0.16 |
| Physical activity score | - | - | 3.6 ± 0.62 | 3.6 ± 0.56 | - |
| **Modeling (mean ± SD)** | | | | | |
| Intake of food score | - | - | 3.9 ± 0.40 | 4.0 ± 0.40 | - |
| Sedentary behavior score | - | - | 2.7 ± 0.52 | 2.6 ± 0.59 | - |
| Physical activity score | - | - | 3.4 ± 0.45 | 3.4 ± 0.45 | - |
| **Setting of rules** | | | | | |
| Breakfast | | | | | 0.13 |
| *No* | 40 (15.9) | 37 (16.6) | 27 (10.7) | 23 (10.4) | |
| *Yes, indulgent* | 27 (10.8) | 14 (6.3) | 17 (6.7) | 17 (7.7) | |
| *Yes, strict* | 184 (73.3) | 172 (77.1) | 208 (82.5) | 181 (81.9) | |
| Snacks | | | | | 0.08 |
| *No* | 52 (20.6) | 53 (23.8) | 40 (16.0) | 36 (16.4) | |
| *Yes, indulgent* | 80 (31.7) | 70 (31.4) | 108 (43.2) | 89 (40.5) | |
| *Yes, strict* | 120 (47.6) | 100 (44.8) | 102 (40.8) | 95 (43.2) | |
| Vegetables | | | | | 0.13 |
| *No* | 51 (20.5) | 46 (20.6) | 33 (13.1) | 24 (10.9) | |

*(Continued)*

**Table 2.** (Continued)

| | According to the children | | According to the parents | | Agreement among answers parent-child |
| --- | --- | --- | --- | --- | --- |
| | Intervention N (%) | Control N (%) | Intervention N (%) | Control N (%) | IRR |
| *Yes, indulgent* | 50 (20.1) | 38 (17.0) | 69 (27.5) | 63 (28.6) | |
| *Yes, strict* | 148 (59.4) | 139 (62.3) | 149 (59.4) | 133 (60.5) | |
| Fruit | | | | | 0.14 |
| *No* | 95 (37.8) | 97 (43.9) | 60 (23.8) | 66 (30.0) | |
| *Yes, indulgent* | 65 (25.9) | 47 (21.3) | 85 (33.7) | 62 (28.2) | |
| *Yes, strict* | 91 (36.3) | 77 (34.8) | 107 (42.5) | 92 (41.8) | |
| SSBs | | | | | 0.16 |
| *No* | 82 (32.7) | 82 (36.9) | 52 (20.8) | 46 (20.9) | |
| *Yes, indulgent* | 84 (33.5) | 54 (24.3) | 76 (30.4) | 67 (30.5) | |
| *Yes, strict* | 85 (33.9) | 86 (38.7) | 122 (48.8) | 107 (48.6) | |
| Watching television | | | | | 0.18 |
| *No* | 111 (44.6) | 107 (48.0) | 85 (34.0) | 79 (35.7) | |
| *Yes, indulgent* | 84 (33.7) | 65 (29.1) | 109 (43.6) | 107 (48.4) | |
| *Yes, strict* | 54 (21.7) | 51 (22.9) | 56 (22.4) | 35 (15.8) | |
| Using the computer | | | | | 0.17 |
| *No* | 99 (39.3) | 77 (34.5) | 70 (28.1) | 58 (26.2) | |
| *Yes, indulgent* | 64 (25.4) | 63 (28.3) | 114 (45.8) | 107 (48.4) | |
| *Yes, strict* | 89 (35.3) | 83 (37.2) | 65 (26.1) | 56 (25.3) | |
| Playing outside | | | | | -0.02 |
| *No* | 193 (76.9) | 163 (73.1) | 158 (63.2) | 141 (63.8) | |
| *Yes, indulgent* | 34 (13.5) | 36 (16.1) | 79 (31.6) | 69 (31.2) | |
| *Yes, strict* | 24 (9.6) | 24 (10.8) | 13 (5.2) | 11 (5.0) | |
| **Parental self-efficacy (mean ± SD)** | | | | | |
| Parenting Sense of Competence | - | - | 76.8 ± 8.2 | 76.9 ± 8.8 | - |

Unless indicated otherwise, data are presented as N (%).

* p≤0.05;

# p≤0.01;

^ p≤.001.

## Process evaluation

In the intention-to-treat approach, 171 of the 252 parent-child dyads (68%) in the intervention condition actually started the e-learning program; 109 of these 171 parents (64%) completed 2 or more modules, while 85 of these parents (50%) completed all 5 modules of the e-learning program (dose received). Of the 81 parent-child dyads in the intervention condition who did not start the e-learning program, 18 (22%) did not have access to the e-learning program due to an incorrect e-mail address (see Fig 2 and Table 5 in S5 Appendix). Failing to start using the e-learning program was more common among non-Caucasian parents compared to Caucasian parents (*p* = 0.009) and among parents with a low level of education compared to parents with a medium or high level of education (*p* = 0.029). In contrast, we found no significant differences between the parents who did not start the e-learning program and the parents who completed 2 or more modules with respect to other demographics or weight-related characteristics (see Table 6 in S5 Appendix). Moreover, the significant differences noted above were not

**Table 3. Estimated marginal means per condition and measurement point and time by condition interactions for continuous outcomes (Intention-to-treat).**

| | | According to children | | | | | According to parents | | | | |
|---|---|---|---|---|---|---|---|---|---|---|---|
| | | T0 | T1 | T2 | Time x condition | | T0 | T1 | T2 | Time x condition | |
| | Condition | M (SE) | M (SE) | M (SE) | p-value | Adj. p-value | M (SE) | M (SE) | M (SE) | p-value | Adj. p-value |
| **Dietary behavior (days/week)** | | | | | | | | | | | |
| Breakfast | CON | 6.54 (0.13) | 6.66 (0.13) | 6.55 (0.14) | 0.875 | 1.000 | 6.54 (0.13) | 6.66 (0.13) | 6.55 (0.14) | 0.545 | 1.000 |
| | INT | 6.60 (0.12) | 6.59 (0.12) | 6.59 (0.13) | | | 6.60 (0.12) | 6.59 (0.12) | 6.59 (0.13) | | |
| Vegetables | CON | 5.68 (0.09) | 5.71 (0.09) | 5.78 (0.09) | 0.643 | 1.000 | 5.51 (0.08) | 5.44 (0.08) | 5.57 (0.08) | 0.631 | 1.000 |
| | INT | 5.59 (0.09) | 5.68 (0.09) | 5.64 (0.09) | | | 5.61 (0.08) | 5.71 (0.08) | 5.70 (0.08) | | |
| Fruit | CON | 5.30 (0.20) | 5.34 (0.20) | 5.10 (0.21) | 0.642 | 1.000 | 5.51 (0.08) | 5.44 (0.08) | 5.57 (0.08) | 0.446 | 1.000 |
| | INT | 5.20 (0.19) | 5.28 (0.19) | 4.92 (0.19) | | | 5.61 (0.08) | 5.71 (0.08) | 5.70 (0.08) | | |
| **Dietary behavior (amount/week)** | | | | | | | | | | | |
| Serving spoons of | CON | 12.6 (0.42) | 11.7 (0.40) | 11.7 (0.39) | 0.856 | 1.000 | 10.0 (0.33) | 10.1 (0.35) | 10.5 (0.35) | 0.509 | 1.000 |
| vegetables | INT | 12.5 (0.40) | 11.9 (0.38) | 11.6 (0.37) | | | 10.1 (0.32) | 10.4 (0.35) | 10.8 (0.34) | | |
| Fruit portions | CON | 15.3 (1.02) | 15.6 (0.90) | 14.2 (0.84) | 0.479 | 1.000 | 11.3 (0.70) | 14.7 (0.72) | 14.1 (0.71) | 0.138 | 1.000 |
| | INT | 15.7 (0.96) | 15.5 (0.84) | 14.1 (0.77) | | | 11.6 (0.64) | 15.6 (0.68) | 15.2 (0.67) | | |
| Glasses of SSBs | CON | 14.9 (0.66) | 15.3 (0.62) | 14.0 (0.61) | 0.093 | 1.000 | 23.0 (1.05) | 21.4 (1.02) | 20.1 (0.96) | 0.443 | 1.000 |
| | INT | 15.6 (0.62) | 15.4 (0.59) | 14.0 (0.59) | | | 23.8 (0.99) | 20.0 (1.01) | 22.0 (0.920 | | |
| **Sedentary behavior** | | | | | | | | | | | |
| Screen time | CON | 796 (63.8) | 704 (58.8) | 797 (62.9) | 0.572 | 1.000 | 996 (41.8) | 852 (43.4) | 1001 (42.8) | 0.768 | 1.000 |
| (min/week) | INT | 707 (58.6) | 701 (54.2) | 741 (58.5) | | | 975 (38.5) | 904 (41.7) | 965 (40.5) | | |
| **Physical activity** | | | | | | | | | | | |
| Playing outside | CON | 630 (44.0) | 654 (44.1) | 443 (44.2) | 0.145 | 1.000 | 489 (29.8) | 576 (31.0) | 395 (30.6) | 0.646 | 1.000 |
| (min/week) | INT | 609 (40.9) | 682 (41.1) | 483 (41.5) | | | 497 (27.4) | 611 (29.8) | 385 (29.0) | | |
| Physical active | CON | 1024 (78.5) | 1060 (71.5) | 982 (69.7) | 0.688 | 1.000 | 903 (33.2) | 1026 (34.8) | 914 (34.3) | 0.041 | 0.820 |
| (min/week) | INT | 1075 (74.6) | 1161 (67.8) | 1006 (65.8) | | | 953 (30.6) | 1064 (34.0) | 875 (32.7) | | |
| **General parenting** | | | | | | | | | | | |
| Involvement | CON | - | - | - | - | - | 12.0 (0.23) | 12.4 (0.25) | 12.3 (0.24) | 0.082 | 1.000 |
| | INT | - | - | - | - | - | 12.6 (0.22) | 12.3 (0.25) | 12.3 (0.24) | | |
| Strictness | CON | - | - | - | - | - | 7.13 (0.27) | 6.84 (0.29) | 6.66 (0.28) | 0.952 | 1.000 |
| | INT | - | - | - | - | - | 7.23 (0.26) | 6.69 (0.29) | 6.79 (0.28) | | |
| **Parental feeding style** | | | | | | | | | | | |
| Control over eating | CON | 3.65 (0.05) | 3.73 (0.05) | 3.71 (0.05) | 0.299 | 1.000 | 4.14 (0.03) | 4.08 (0.04) | 4.01 (0.04) | 0.966 | 1.000 |
| eating | INT | 3.64 (0.05) | 3.64 (0.04) | 3.63 (0.04) | | | 4.12 (0.03) | 4.07 (0.04) | 3.99 (0.03) | | |
| Emotional feeding | CON | 1.55 (0.05) | 1.49 (0.04) | 1.42 (0.04) | 0.016 | 0.224 | 1.35 (0.03) | 1.37 (0.03) | 1.31 (0.03) | 0.299 | 1.000 |
| | INT | 1.59 (0.04) | 1.40 (0.04) | 1.32 (0.04) | | | 1.34 (0.03) | 1.34 (0.03) | 1.34 (0.03) | | |
| Encouragement | CON | 2.85 (0.06) | 2.88 (0.06) | 2.86 (0.06) | 0.217 | 1.000 | 3.64 (0.04) | 3.67 (0.04) | 3.62 (0.05) | 0.979 | 1.000 |
| to eat | INT | 2.86 (0.06) | 2.72 (0.06) | 2.78 (0.06) | | | 3.74 (0.04) | 3.78 (0.04) | 3.73 (0.05) | | |
| Instrumental | CON | 1.70 (0.05) | 1.55 (0.04) | 1.48 (0.04) | 0.545 | 1.000 | 1.50 (0.04) | 1.53 (0.04) | 1.43 (0.04) | 0.464 | 1.000 |
| feeding | INT | 1.71 (0.05) | 1.55 (0.04) | 1.45 (0.04) | | | 1.53 (0.04) | 1.48 (0.04) | 1.43 (0.04) | | |
| Monitoring | CON | 3.63 (0.07) | 3.58 (0.08) | 3.47 (0.08) | 0.061 | 0.793 | 3.67 (0.04) | 3.61 (0.04) | 3.51 (0.04) | 0.061 | 1.000 |
| physical activity | INT | 3.60 (0.07) | 3.66 (0.07) | 3.53 (0.07) | | | 3.62 (0.04) | 3.68 (0.04) | 3.55 (0.04) | | |
| **Modeling** | | | | | | | | | | | |
| Intake of food | CON | - | - | - | - | - | 3.96 (0.03) | 3.93 (0.03) | 3.95 (0.03) | 0.098 | 1.000 |

*(Continued)*

**Table 3.** (Continued)

| | Condition | According to children | | | | | According to parents | | | | |
|---|---|---|---|---|---|---|---|---|---|---|---|
| | | T0 | T1 | T2 | Time x condition | | T0 | T1 | T2 | Time x condition | |
| | Condition | M (SE) | M (SE) | M (SE) | p-value | Adj. p-value | M (SE) | M (SE) | M (SE) | p-value | Adj. p-value |
| | INT | - | - | - | - | - | 3.92 (0.03) | 3.91 (0.03) | 3.97 (0.03) | | |
| Sedentary | CON | - | - | - | - | - | 2.62 (0.04) | 2.68 (0.04) | 2.64 (0.04) | 0.744 | 1.000 |
| behavior | INT | - | - | - | - | - | 2.66 (0.04) | 2.69 (0.04) | 2.70 (0.04) | | |
| Physical activity | CON | - | - | - | - | - | 3.43 (0.03) | 3.49 (0.03) | 3.43 (0.03) | 0.310 | 1.000 |
| | INT | - | - | - | | | 3.42 (0.03) | 3.49 (0.03) | 3.46 (0.03) | | |
| **Parental self-efficacy** | | | | | | | | | | | |
| Parenting sense | CON | - | - | - | - | - | 76.9 (0.62) | 76.6 (0.64) | 77.8 (0.63) | 0.735 | 1.000 |
| of competence | INT | - | - | - | - | - | 76.8 (0.59) | 76.5 (0.63) | 77.4 (0.62) | | |

CON = Control condition; INT = Intervention condition.

found in the completers-only framework. Finally, the parents' mean (±SD) satisfaction with the e-learning program (on a 10-point scale) was 7.0±1.1.

## Discussion

The aim of this study was to determine the added value of including our e-learning parenting program entitled "Making a healthy deal with your child" in addition to the existing school-based overweight prevention program "Scoring for Health". The intention-to-treat analyses and completers only sensitivity analyses revealed no significant differences in the changes over time between the intervention and the control condition with respect to the EBRBs of the child, or to parenting dimensions. We initially found that parents in the intervention condition used an emotional feeding style less often and more frequently set rules regarding using the computer; however, these effects were no longer significant after correcting for multiple testing. In addition, our evaluations of the program revealed that the e-learning program was well received by the parents, and parents were generally satisfied with it.

Our results are not in line with the results reported by Gerards et al., who studied the effectiveness of the parenting program "Lifestyle Triple P" [78] who found some positive effects with respect to EBRBs of children; however, it is important to note that Gerards et al. study included younger children (4–8 years of age) than our study (9–13 years of age), and all children in their study were overweight or obese, and their program included ten face-to-face group meetings and four individual telephone interviews.

Our results regarding the failure to find intervention effects on parenting skills are also not in line with the results reported by Leijten et al. [79], who examined the effectiveness of the parental program entitled "Talking with your child", and of which we used the theory for the development of our e-learning "Making a healthy deal with your child", and a meta-analysis by Nieuwboer et al. who found that knowledge can be increased, and attitudinal and behavioral aspects of parenting can be influenced by online programs, leading to the conclusion that the Internet is not only a source of information, but can also be an instrument for parental support and training [80].

Our finding that the e-learning program had no apparent effects can lead to different conclusions. In fact, using the optimal intervention requires combining the appropriate target/objectives for an intervention, matching the intervention with a well-chosen target group, and

**Table 4. Observed probabilities per condition and measurement point and time by condition interactions for dichotomized outcomes.**

| | | According to children | | | | | According to parents | | | | |
|---|---|---|---|---|---|---|---|---|---|---|---|
| | | T0 | T1 | T2 | Time x condition | | T0 | T1 | T2 | Time x condition | |
| | Condition | Prob (SD) | Prob (SD) | Prob (SD) | p-value | Adj. p-value | Prob (SD) | Prob (SD) | Prob (SD) | p-value | Adj. p-value |
| **Dietary behavior (daily)** | | | | | | | | | | | |
| Breakfast | CON | 0.84 (0.02) | 0.89 (0.02) | 0.85 (0.02) | 0.733 | 1.000 | 0.84 (0.02) | 0.86 (0.03) | 0.88 (0.02) | 0.780 | 1.000 |
| | INT | 0.83 (0.02) | 0.84 (0.02) | 0.85 (0.02) | | | 0.86 (0.02) | 0.91 (0.02) | 0.90 (0.02) | | |
| Vegetables | CON | 0.30 (0.03) | 0.30 (0.03) | 0.33 (0.03) | 0.653 | 1.000 | 0.20 (0.03) | 0.20 (0.03) | 0.26 (0.03) | 0.839 | 1.000 |
| | INT | 0.28 (0.03) | 0.30 (0.03) | 0.28 (0.03) | | | 0.24 (0.03) | 0.29 (0.04) | 0.33 (0.03) | | |
| > 2 portions of | CON | 0.18 (0.03) | 0.15 (0.02) | 0.15 (0.02) | 0.358 | 1.000 | 0.04 (0.01) | 0.13 (0.03) | 0.09 (0.02) | 0.318 | 1.000 |
| fruit | INT | 0.20 (0.03) | 0.19 (0.03) | 0.12 (0.02) | | | 0.06 (0.01) | 0.12 (0.02) | 0.12 (0.02) | | |
| <2 glasses of SSBs | CON | 0.25 (0.03) | 0.25 (0.03) | 0.30 (0.03) | 0.383 | 1.000 | 0.31 (0.03) | 0.31 (0.03) | 0.32 (0.03) | 0.212 | 1.000 |
| | INT | 0.21 (0.03) | 0.24 (0.03) | 0.30 (0.03) | | | 0.23 (0.03) | 0.31 (0.04) | 0.29 (0.03) | | |
| **Sedentary behavior** | | | | | | | | | | | |
| Daily ≤ 120 min | CON | 0.65 (0.03) | 0.70 (0.03) | 0.62 (0.03) | 0.991 | 1.000 | 0.37 (0.03) | 0.49 (0.04) | 0.36 (0.03) | 0.564 | 1.000 |
| screen time | INT | 0.68 (0.03) | 0.68 (0.03) | 0.67 (0.03) | | | 0.37 (0.03) | 0.45 (0.04) | 0.37 (0.03) | | |
| **Physical activity** | | | | | | | | | | | |
| Dutch standard | CON | 0.63 (0.03) | 0.69 (0.03) | 0.60 (0.03) | 0.587 | 1.000 | 0.66 (0.03) | 0.72 (0.03) | 0.69 (0.03) | 0.024 | 0.528 |
| exercise | INT | 0.78 (0.03) | 0.82 (0.03) | 0.75 (0.03) | | | 0.80 (0.03) | 0.83 (0.03) | 0.77 (0.03) | | |
| Play outside >1 | CON | 0.31 (0.03) | 0.36 (0.03) | 0.16 (0.03) | 0.511 | 1.000 | 0.23 (0.03) | 0.33 (0.04) | 0.19 (0.03) | 0.979 | 1.000 |
| hour daily | INT | 0.29 (0.03) | 0.37 (0.03) | 0.18 (0.03) | | | 0.25 (0.03) | 0.43 (0.04) | 0.19 (0.03) | | |
| **General parenting** | | | | | | | | | | | |
| Authoritative | CON | - | - | - | - | | 0.29 (0.03) | 0.31 (0.03) | 0.31 (0.03) | 0.677 | 1.000 |
| parenting style | INT | - | - | - | | | 0.33 (0.03) | 0.29 (0.03) | 0.33 (0.03) | | |
| **Setting of rules** | | | | | | | | | | | |
| Breakfast | | | | | | | | | | | |
| No | CON | 0.17 (0.02) | 0.19 (0.03) | 0.14 (0.02) | 0.187 | 1.000 | 0.10 (0.02) | 0.07 (0.02) | 0.10 (0.02) | 0.973 | 1.000 |
| | INT | 0.16 (0.02) | 0.21 (0.03) | 0.19 (0.03) | | | 0.11 (0.02) | 0.14 (0.03) | 0.10 (0.02) | | |
| Indulgent | CON | 0.06 (0.02) | 0.08 (0.02 | 0.11 (0.02) | | | 0.08 (0.02) | 0.07 (0.02) | 0.05 (0.02) | | |
| | INT | 0.11 (0.02) | 0.13 (0.02) | 0.12 (0.02) | | | 0.07 (0.02) | 0.05 (0.02) | 0.07 (0.02) | | |
| Strict | CON | 0.77 (0.03) | 0.74 (0.03) | 0.74 (0.03) | 0.776 | 1.000 | 0.82 (0.03) | 0.86 (0.03) | 0.81 (0.03) | 0.642 | 1.000 |
| | INT | 0.73 (0.03) | 0.66 (0.03) | 0.69 (0.03) | | | 0.83 (0.02) | 0.81 (0.03) | 0.84 (0.03) | | |
| Snacks | | | | | | | | | | | |
| No | CON | 0.24 (0.03) | 0.19 (0.03) | 0.22 (0.03) | 0.161 | 1.000 | 0.16 (0.03) | 0.11 (0.02) | 0.14 (0.02) | 0.753 | 1.000 |
| | INT | 0.21 (0.03) | 0.20 (0.03) | 0.26 (0.03) | | | 0.16 (0.03) | 0.16 (0.03) | 0.14 (0.02) | | |
| Indulgent | CON | 0.31 (0.03) | 0.40 (0.03) | 0.35 (0.03) | | | 0.40 (0.03) | 0.43 (0.04) | 0.44 (0.04) | | |
| | INT | 0.32 (0.03) | 0.39 (0.03) | 0.36 (0.03) | | | 0.43 (0.03) | 0.47 (0.04) | 0.45(0.04) | | |
| Strict | CON | 0.45 (0.03) | 0.40 (0.03) | 0.43 (0.03) | 0.182 | 1.000 | 0.43 (0.03) | 0.45 (0.04) | 0.42 (0.04) | 0.809 | 1.000 |
| | INT | 0.48 (0.03) | 0.41 (0.03) | 0.38 (0.03) | | | 0.41 (0.03) | 0.37 (0.04) | 0.41 (0.03) | | |
| Vegetables | | | | | | | | | | | |
| No | CON | 0.21 (0.03) | 0.22 (0.03) | 0.21 (0.03) | 0.235 | 1.000 | 0.11 (0.02) | 0.08 (0.02) | 0.13 (0.02) | 0.580 | 1.000 |
| | INT | 0.20 (0.03) | 0.29 (0.03) | 0.27 (0.03) | | | 0.13 (0.02) | 0.15 (0.03) | 0.11 (0.02) | | |
| Indulgent | CON | 0.17 (0.03) | 0.20 (0.03) | 0.21 (0.03) | | | 0.29 (0.03) | 0.31 (0.03) | 0.28 (0.03) | | |
| | INT | 0.20 (0.03) | 0.24 (0.03) | 0.24 (0.03) | | | 0.27 (0.03) | 0.28 (0.03) | 0.23 (0.03) | | |
| Strict | CON | 0.62 (0.03) | 0.59 (0.03) | 0.59 (0.03) | 0.249 | 1.000 | 0.60 (0.03) | 0.61 (0.04) | 0.59 (0.04) | 0.378 | 1.000 |
| | INT | 0.59 (0.03) | 0.48 (0.03) | 0.49 (0.03) | | | 0.59 (0.03) | 0.57 (0.04) | 0.66 (0.03) | | |
| Fruit | | | | | | | | | | | |
| No | CON | 0.44 (0.03) | 0.43 (0.03) | 0.45 (0.03) | 0.046 | 1.000 | 0.30 (0.03) | 0.22 (0.03) | 0.28 (0.03) | 0.863 | 1.000 |
| | INT | 0.38 (0.03) | 0.45 (0.03) | 0.49 (0.03) | | | 0.24 (0.03) | 0.23 (0.03) | 0.21 (0.03) | | |

*(Continued)*

**Table 4.** (Continued)

| | | According to children | | | | | According to parents | | | | |
| | | T0 | T1 | T2 | Time x condition | | T0 | T1 | T2 | Time x condition | |
| | Condition | Prob (SD) | Prob (SD) | Prob (SD) | p-value | Adj. p-value | Prob (SD) | Prob (SD) | Prob (SD) | p-value | Adj. p-value |
|---|---|---|---|---|---|---|---|---|---|---|---|
| Indulgent | CON | 0.21 (0.03) | 0.24 (0.03) | 0.22 (0.03) | | | 0.28 (0.03) | 0.37 (0.04) | 0.33 (0.03) | | |
| | INT | 0.26 (0.03) | 0.26 (0.03) | 0.24 (0.03) | | | 0.34 (0.03) | 0.31 (0.04) | 0.35 (0.03) | | |
| Strict | CON | 0.35 (0.03) | 0.33 (0.03) | 0.32 (0.03) | 0.194 | 1.000 | 0.42 (0.03) | 0.41 (0.04) | 0.39 (0.04) | 0.534 | 1.000 |
| | INT | 0.36 (0.03) | 0.29 (0.03) | 0.27 (0.03) | | | 0.42 (0.03) | 0.46 (0.04) | 0.44 (0.03) | | |
| SSBs | | | | | | | | | | | |
| No | CON | 0.37 (0.03) | 0.35 (0.03) | 0.36 (0.03) | 0.934 | 1.000 | 0.21 (0.03) | 0.18 (0.03) | 0.14 (0.03) | 0.370 | 1.000 |
| | INT | 0.33 (0.03) | 0.29 (0.03) | 0.32 (0.03) | | | 0.21 (0.03) | 0.19 (0.03) | 0.18 (0.03) | | |
| Indulgent | CON | 0.24(0.03) | 0.22 (0.03) | 0.31 (0.03) | | | 0.30 (0.03) | 0.30 (0.03) | 0.41 (0.04) | | |
| | INT | 0.33 (0.03) | 0.36 (0.03) | 0.35 (0.03) | | | 0.30 (0.03) | 0.28 (0.03) | 0.38 (0.03) | | |
| Strict | CON | 0.39 (0.03) | 0.44 (0.03) | 0.33 (0.03) | 0.495 | 1.000 | 0.49 (0.03) | 0.51 (0.04) | 0.45 (0.04) | 0.780 | 1.000 |
| | INT | 0.34(0.03) | 0.35 (0.03) | 0.32 (0.03) | | | 0.49 (0.03) | 0.53 (0.04) | 0.44 (0.04) | | |
| Watching television | | | | | | | | | | | |
| No | CON | 0.48 (0.03) | 0.49 (0.03) | 0.48 (0.03) | 0.866 | 1.000 | 0.36 (0.03) | 0.30 (0.03) | 0.35 (0.03) | 0.056 | 1.000 |
| | INT | 0.45 (0.03) | 0.47 0.03) | 0.43 (0.03) | | | 0.34 (0.03) | 0.23 (0.03) | 0.25 (0.03) | | |
| Indulgent | CON | 0.29 (0.03) | 0.26 (0.03) | 0.35 (0.03) | | | 0.48 (0.03) | 0.54 (0.04) | 0.48 (0.04) | | |
| | INT | 0.34 (0.03) | 0.28 (0.03) | 0.35 (0.03) | | | 0.44 (0.03) | 0.53 (0.04) | 0.50 (0.04) | | |
| Strict | CON | 0.23 (0.03) | 0.25 (0.03) | 0.17 (0.03) | 0.238 | 1.000 | 0.16 (0.02) | 0.16 (0.03) | 0.17 (0.03) | 0.964 | 1.000 |
| | INT | 0.22 (0.03) | 0.25 (0.03) | 0.22 (0.03) | | | 0.22 (0.03) | 0.23 (0.03) | 0.24(0.03) | | |
| Using the computer | | | | | | | | | | | |
| No | CON | 0.35 (0.03) | 0.34 (0.03) | 0.41 (0.03) | 0.190 | 1.000 | 0.26 (0.03) | 0.24 (0.03) | 0.26 (0.03) | 0.021 | 0.483 |
| | INT | 0.39 (0.03) | 0.37 (0.03) | 0.40 (0.03) | | | 0.28 (0.03) | 0.19 (0.03) | 0.22 (0.03) | | |
| Indulgent | CON | 0.28 (0.03) | 0.36 (0.03) | 0.32 (0.03) | | | 0.48 (0.03) | 0.50 (0.04) | 0.50 (0.04) | | |
| | INT | 0.25 (0.03) | 0.29 (0.03) | 0.30 (0.03) | | | 0.46 (0.03) | 0.45 (0.04) | 0.44 (0.04) | | |
| Strict | CON | 0.37 (0.03) | 0.30 (0.03) | 0.27 (0.03) | 0.209 | 1.000 | 0.25 (0.03) | 0.26 (0.03) | 0.24 (0.03) | 0.094 | 1.000 |
| | INT | 0.35 (0.03) | 0.34 (0.03) | 0.31 (0.03) | | | 0.26 (0.03) | 0.35 (0.04) | 0.34 (0.03) | | |
| Playing outside | | | | | | | | | | | |
| No | CON | 0.73 (0.03) | 0.75 (0.03) | 0.77 (0.03) | 0.817 | 1.000 | 0.64 (0.03) | 0.54 (0.04) | 0.63 (0.03) | 0.211 | 1.000 |
| | INT | 0.77 (0.03) | 0.83 (0.03) | 0.80 (0.03) | | | 0.63 (0.03) | 0.50 (0.04) | 0.56 (0.04) | | |
| Indulgent | CON | 0.16 (0.02) | 0.14 (0.02) | 0.14 (0.02) | | | 0.31 (0.03) | 0.42 (0.04) | 0.32 (0.03) | | |
| | INT | 0.14 (0.02) | 0.12(0.02) | 0.14 (0.02) | | | 0.32 (0.03) | 0.39 (0.04) | 0.36 (0.03) | | |
| Strict | CON | 0.11 (0.02) | 0.11 (0.02) | 0.09 (0.02) | 0.701 | 1.000 | 0.05(0.01) | 0.04 (0.02) | 0.05 (0.02) | 0.045 | 0.945 |
| | INT | 0.10 (0.02) | 0.06 (0.02) | 0.07 (0.02) | | | 0.05 (0.01) | 0.10 (0.02) | 0.09 (0.02) | | |

The multicategory variables for parental EBRB rules were dichotomized in order to compare between "no rules" and "indulgent/strict rules" and between "strict rules" and "no/indulgent rules". Therefore 2 p-values for each parental EBRB rules are presented.

identifying the ideal intervention to use. This process is also known as selecting the ideal 'target-target group-intervention combination' [81]. This can have several practical implications. First, the target may need to be adjusted as the measurements are affected by so many variables and factors that, when combined with the general target group to which the intervention was tested, any effects may be averaged out. Second, the target group to which we offered the intervention may have been too general, and/or had insufficient urgency regarding the need to change their parenting skills. For example, Gerards et al. included only parents of children with overweight and/or obesity [78], while Leijten et al. included only parents who experienced parenting difficulties [79]. Because this was not the case in our intervention, it may be possible

that our intervention should be offered more targeted parental groups, for example, parents of children with unhealthy EBRBs and/or with overweight. However, systematic reviews by both Bleich et al. and Wang et al. showed that the most promising interventions are delivered in school settings that include a home component [82,83]. In addition, based on their review, Tomayko et al.'s findings support including a parental component in school-based treatment and prevention programs designed to improve the child's weight and/or weight status outcome [23]. Lastly, our findings indicate that the program has room for improvement, as the intervention was not used intensively enough. The intensity at which the parents in our study were exposed to the intervention also warrants discussion. We found that 64% of parents in the intervention condition who started the e-learning program completed 2 or more modules, with 50% of parents completing all 5 modules (the total delivery dose). However, if we consider all 252 parent-child dyads in the intervention condition (i.e., both the group of parents who started the program and the group of parents who never started the program), only 43% and 34% of all parents assigned to the intervention condition completed 2 or more modules or all 5 modules, respectively. Following 2 or more of the 5 modules is the minimum requirement to be considered sufficiently exposed to the e-learning program. Thus, in our study only 43% of the total intervention condition was sufficiently exposed to the intervention. Therefore, the relatively low percentage of parents who were sufficiently exposed to the intervention may explain why we found no apparent difference between the intervention and control conditions, making it difficult to determine whether the intervention was actually effective given the high percentage of parents (57%) who were not sufficiently exposed to the intervention. Nevertheless, the parents' evaluations showed that the program was generally well received. For example parents were enthusiastic with respect to the simple use of the e-learning program and the fact that the program can be followed in their home at a time that suits them best. In addition, the theory, content, and structure of this program were derived from an online course entitled "Talking with your child", which was based on other successful behavioral themes [43]. In addition, we used a relatively minimal engagement strategy. Our goal was to develop a self-guided web-based parenting program that can be added easily to existing overweight-prevention programs such as existing school-based/population-based interventions, thereby maximizing integration and minimizing costs. In one respect, this minimal-engagement strategy was successful, as 68% of parents started the intervention, with 64% of them completing at least 2 of the 5 modules, and 50% completing all 5 modules. Unfortunately, despite this relatively high level of involvement we found that the intervention had no apparent effects on either parenting skills or the children's EBRBs. Using additional strategies to increase engagement in addition to sending two email reminders may have increased the percentage of parents who participated in the intervention, thus increasing the likelihood of revealing an intervention effect. In their supportive accountability model, Mohr et al. argue that human support increases adherence by increasing accountability to a coach who is seen as trustworthy, benevolent, and having expertise, moderated by motivational factors such as the communication media used (e.g., face-to-face, telephone, or email) [84]. However, we expect that parents of children in a high-risk group will be more willing to change their child's EBRBs, feel a stronger sense of urgency to change their parenting skills, and would therefore be more motivated to use the e-learning program. We would therefore expect a higher percentage of parents to follow the e-learning program, thus revealing intervention effects. Moreover, we expect to find more room for improvement among the high-risk group with respect to improving parenting skills and their child's EBRBs. On the other hand, if we target a high-risk group who are currently under the care of a health-care professional, it should be easier to increase engagement, as a trusted health-care professional is already involved and can provide the parents with face-to-face support, increasing their likelihood of starting—and ideally, completing—the

e-learning program. This healthcare professional can also make follow-up appointments with the parents, and use this appointment also to discuss whether the e-learning program was completed successfully, or—if necessary—encourage them to complete the e-learning program. Due to the parents' positive evaluations and enthusiasm regarding the program's simplicity and flexibility, we believe that the next step is to first select a more appropriate (i.e. high-risk) target group and conduct further research in that context.

### Strengths and limitations

The strengths of our study include its cluster RCT design, the relatively large sample size, and the fact that we collected data at baseline, 5 months, and 12 months, which enabled us to analyze the relatively short-term and long-term effects of the program. Furthermore, our insights into which parents began the e-learning program (taking into account the child's socio-demographic characteristics and weight status), and which parents did not start the program, as well as how many modules the parents completed, provided valuable information regarding which parents are—and just as important, which parents are not—reached using this intervention, thus helping to maximize the scope of the e-learning program.

The e-learning program itself also has several strengths. First, the program is theory-driven and based on difficult situations experienced in daily life by most parents. Second, the program consists of multiple components, including video fragments, a six-step problem-solving model, assignments, and feedback, which is important given that reports have shown that multi-component programs can yield more effects than single-component programs [22,85]. Third, because the e-learning program is web-based, parents can follow it in the comfort of their own home, in their own time and at their own pace; moreover, the program is not complex or time-consuming. Finally, the children's baseline EBRBs and the prevalence of overweight and obesity in our sample are consistent with children in the general Dutch population; therefore, we believe that our sample provided a reasonably good representation of parents and their children, and the results are not likely biased due to selective participation [86].

Despite its strengths, this study has some limitations that warrant discussion. First, we cannot rule out the possibility that the more highly motivated parents completed the e-learning program, which may limit our ability to generalize our results to all parents. Second, using a within-school design (as opposed to a between-school design) may have introduced contamination effects between the intervention and control conditions. To minimize this possibility, the same brochure was given to the parents in the intervention condition and the control condition, while only the parents in the intervention condition received a personal login code in order to start the e-learning program. Third, the information regarding the behaviors of the children and parents was obtained entirely using self-reporting questionnaires completed by the children and parents, which may have resulting in over-reporting and/or under-reporting due social desirability and/or recall bias and may contribute to unreliability of the reported data [87]. To minimize social desirability and optimize measurement validity, we stressed full confidentiality (i.e., anonymity) to our participants. To minimize recall bias, the interval between the events being recalled and the questionnaire was relatively short (participants were asked to recall events from the past month or week), which likely also increased the self-reporting reliability. To increase reliability it may have been better to use accelerometers to report physical activity; however, for determining whether the children meet or did not meet the Dutch standard for healthy exercise, a questionnaire may be appropriate [88]. Fourth, the amount of time parents needed to complete the program is relatively short, with 5 modules of approximately 30 minutes each; thus, program was not necessarily expected to have a robust effect.

## Practical implications and conclusions

Despite the parent's overall satisfaction regarding the e-learning program and the ease with which our e-learning program can be integrated into the "Scoring for health" program, we found no significant differences between the intervention and control condition. Nevertheless, our results provide new insights into the factors that both contribute to and foster intervention effects, thus leading to further improvements in our e-learning program and possibly other universally implemented programs for overweight prevention. Demonstrating effects of our e-learning program was potentially complicated by the fact that we studied the effects in children in a school-based setting, in which many children may already have engaged in healthy EBRBs. Thus, the e-learning program may have beneficial effects if added to an existing over-weight-prevention program currently being offered to high-risk groups, for example in the Youth Health Care or in a hospital setting, in which we expect the parents will feel a stronger sense of urgency and therefore be more motivated to use the e-learning program. Future research should test the effectiveness of this e-learning program specifically in high risk groups, such as parents of children with unhealthy EBRBs, and/or with overweight, thus providing more information regarding the benefits of this program in this specific target population.

## Supporting information

**S1 Appendix. Overview of parenting scale.**
(DOCX)

**S2 Appendix. Demographic and weight-related characteristics of study population (Completers-only).**
(DOCX)

**S3 Appendix. Baseline characteristics of children's EBRB and parenting dimensions (Completers only).**
(DOCX)

**S4 Appendix. Effects of the e-learning on the EBRBs of the child and parenting dimensions (Completers-only).**
(DOCX)

**S5 Appendix. Process evaluation.**
(DOCX)

## Acknowledgments

This study is part of the Dutch project CIAO (Consortium Integrated Approach Overweight) in which several studies are investigating elements of a coherent integrated multi-sector approach based on the principles of the French program *Ensemble Prévenons l'Obésité Des Enfants* (EPODE; in English, "Together Let's Prevent Childhood Obesity") [89]. We thank Hans Bör of the Radboud University Medical Center for help with the power calculation. In addition, we would like to thank all the interns who assisted in this research.

## Author Contributions

**Conceptualization:** Emilie L. M. Ruiter, Gerard R. M. Molleman, Koos van der Velden, Rutger C. M. E. Engels, Gerdine A. J. Fransen.

**Formal analysis:** Emilie L. M. Ruiter, Marloes Kleinjan, Jannis T. Kraiss, Peter M. ten Klooster.

**Methodology:** Gerard R. M. Molleman, Koos van der Velden, Rutger C. M. E. Engels, Gerdine A. J. Fransen.

**Project administration:** Emilie L. M. Ruiter.

**Supervision:** Gerard R. M. Molleman, Marloes Kleinjan, Koos van der Velden, Rutger C. M. E. Engels, Gerdine A. J. Fransen.

**Writing – original draft:** Emilie L. M. Ruiter.

**Writing – review & editing:** Emilie L. M. Ruiter, Gerard R. M. Molleman, Marloes Kleinjan, Jannis T. Kraiss, Peter M. ten Klooster, Koos van der Velden, Rutger C. M. E. Engels, Gerdine A. J. Fransen.

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
