## [Decision Letter · Decision Letter 0]

21 Oct 2021

PONE-D-21-30893

The effectiveness of a web-based Dutch parenting program to prevent overweight in children 9-13 years of age: results of a two-armed cluster randomized controlled trial

PLOS ONE

Dear Dr. Ruiter,

Thank you for submitting your manuscript to PLOS ONE. After careful consideration, we feel that it has merit but does not fully meet PLOS ONE’s publication criteria as it currently stands. Therefore, we invite you to submit a revised version of the manuscript that addresses the points raised during the review process.

The decision for this paper is a difficult one. Let me start by saying that I believe it is important to publish this paper, its just a difficult situation. On the one hand you have taken the appropriate steps, registering the clinical trial, publishing a study protocol, and then for the most part following that protocol in this paper. On the other hand, I have many of the same concerns that are raised by the statistical reviewer as well as a few others. I think you should know where I’m coming from as an editor, while I’m trained as a social scientist, my primary role is as a methodologist, I’ve been the primary statistical methods person on multiple randomized and group randomized trials, some targeting physical activity and weight loss. I am also a reviewer for major clinical trials grants in the area. If I had reviewed either the grant application or the study protocol paper I would have had reservations about the statistical methods proposed (doing analyses in Mplus with TYPE = COMPLEX, failure to account for multiple testing, dichotomizing outcomes, treating T1 and T2 as separate outcomes, reporting both ITT and completers only analysis) and would have requested major revisions. So, now the question I’m facing is how to reconcile the desire to stay true to the published protocol paper versus moving forward with results that do not meet the standard I would expect from a clinical trial. My overall decision at this point is to request major revisions which try to address these issues. I’ll also note that I held off sending this paper out to content specific reviewers because I wanted to see if the statistics reviewer had the same concerns I did, and because the substantive reviews will likely be better if these other issues are fixed.

Several things seem straight forward to me: 1) the protocol paper did not include per-protocol analyses, I believe that these need to be dropped or if they are retained they need to be done using Complier Average Causal Effects analyses appropriate for cluster randomized trials (there is a paper out there on doing this by Booil Jo), but these are difficult analyses to do and I would want the code and output included so that they could be checked; 2) the protocol paper did not mention how multiple outcomes would be tested, this is really a major problem, you have 13 outcomes multiplied by two different reporters (parents and children) and two different time points, so 52 different analyses for 3 different types of analyses. The trial registration includes 7 primary outcomes, only a little bit better. This needs to be addressed and I’m not sure that you have many options at this point, but the end result is likely that there are no reliable differences between treatment and control groups; 3) Your actual statistical analyses need to be made clear. You simply report that you used Mplus with type=complex, this is not the statistical analysis, it is just the software used. As the statistical reviewer mentions, the models used need to be reported in careful detail with all predictors and outcomes specified, assumptions tested, and variance components reported; 4) the justification for the sample size calculations needs to be made in much more detail including both the ICC assumptions and the effect size assumptions along with references; 5) Missing data and imputation needs to be described in more detail.

Now the difficult things. First of all is your use of Mplus and type=complex to account for the clustering in the data. I’m going to start by nothing that you are wrong about this being multilevel analysis, when you use the type=complex option Mplus simply adjusts the standard errors of your effects for clustering using a sandwich estimator, but this is not a multilevel regression model. The first reason I don’t typically suggest Mplus for multilevel models is that users often don’t understand what is happening, the second is that many Mplus features are not well documented and it really isn’t clear what sandwich estimator is employed. In any case, this should be similar to more typical implementations of GEE. While GEE can be acceptable in GRTs for correcting for clustering, it isn’t very efficient and the approach recommended by the reviewer is clearly the standard. A recent paper by Turner et al (American J of Public Health) discusses some of these issues. If the protocol paper wasn’t published I would feel quite strongly (along with the stats reviewer) that the current analyses should be completely replaced with the conventional multilevel models recommended by the reviewer. However, given the existence of the protocol paper, my suggestion is that the ITT and completers analyses stays as is (with the full model including all covariates and variance parameter estimates described) but that the analyses suggested by the reviewer be included in an appendix and any discrepancies noted. There really shouldn’t be large discrepancies if everything was done right, and my reason for asking for this is that it will greatly increase my confidence that everything was done right.

Second, were covariates included in these analyses? Also, was the analyses the outcome at time T controlling for baseline or was it change scores? If covariates were not included, why not? I agree with the reviewer that baseline values for weight as other variables should have been included, the protocol paper is not clear about how or if this will be done.

Third, I’m not sure that all of your effect sizes are even in the correct direction for the few significant treatment effects. Looks like there were more sweetened beverages consumed in the treatment than control groups? Also, some of the significance tests in the completers only table seem to be off? Oh, the use of superscripts to indicate which effects are significant for which set of analyses doesn’t work at all. You should probably just report the full results of the ITT and then, separately, of the completers-only analyses (again, dropping the protocol analyses).

I’ve noted several other issues. One is that I doubt that substantive reviewers would be impressed with your outcomes for the dietary or physical activity measures. On the trials I’ve been on we have ignored self or parent reported PA and focused exclusively on accelerometer measured PA because of concerns about bias as well as unreliability.  Obviously you can’t change this now, but it needs to be a very important discussion point and the issues with the measures need to be made very clear.

Another issue which is really major is that I do not think you have interpreted these results correctly. You conclude that, “our web-based parenting program yields promising effects…” and when I look at your results the ITT analyses (these should be the gold standard) truly show no effects if you do any meaningful correction for multiple comparisons. Again, it looks to me like some of your results are in the wrong direction which also suggests no overall effects. The completers only analyses looks about the same to me. My best guess looking at these results is that this study has more evidence of no effects than of effects, the only place where you see something is the per-protocol analyses but those results are well known to be biased the way you did them.

My final note is that I disagree with the reviewer about dropping the tables with parameter estimates and standard errors, I would focus more rather than less on them.

We look forward to receiving your revised manuscript.

Kind regards,

Lee Van Horn, PhD

Academic Editor

PLOS ONE

Journal Requirements:

3.  Please provide the original, full-length study protocol, or project proposal, that you submitted to your ethics committee for review and approval prior to study commencement. This document should typically contain a background section, detailed description of the methods used (including information about dates of recruitment, participant characteristics, inclusion/exclusion criteria, detailed descriptions of the protocols/interventions utilized, plans for outcome assessment and data collection/analysis, and references). Please note that we require this to be submitted with your manuscript in both the original language version and a version translated into English.

4.  Thank you for submitting your clinical trial to PLOS ONE and for providing the name of the registry and the registration number. The information in the registry entry suggests that your trial was registered after patient recruitment began. PLOS ONE strongly encourages authors to register all trials before recruiting the first participant in a study.

a) your reasons for your delay in registering this study (after enrolment of participants started);

b) confirmation that all related trials are registered by stating: “The authors confirm that all ongoing and related trials for this drug/intervention are registered

"This study was funded by a grant from the Netherlands Organization for Health Research and Development (ZonMw; project number 505010296015), which played no role in the design of execution of this study, the analysis or interpretation of the data, or the decision to publish the results."

"This study was funded by a grant from the Netherlands Organization for Health Research and Development (ZonMw; project number 505010296015, 200100001). This funder had no role in study design, data collection and analysis, interpretation of the data,  decision to publish the results, or preparation of the manuscript."

6. We note that you have indicated that data from this study are available upon request. PLOS only allows data to be available upon request if there are legal or ethical restrictions on sharing data publicly. For information on unacceptable data access restrictions, please see http://journals.plos.org/plosone/s/data-availability#loc-unacceptable-data-access-restrictions. 

Reviewers' comments:

Reviewer's Responses to Questions

Comments to the Author

1. Is the manuscript technically sound, and do the data support the conclusions?

Reviewer #1: Partly

2. Has the statistical analysis been performed appropriately and rigorously?

Reviewer #1: No

3. Have the authors made all data underlying the findings in their manuscript fully available?

Reviewer #1: Yes

4. Is the manuscript presented in an intelligible fashion and written in standard English?

Reviewer #1: Yes

5. Review Comments to the Author

Reviewer #1: Sample size: why the intra class correlation is set at a low level of 0.05?

Missing data: what method was used for imputation? There are discrepancies between ITT and per-protocol analysis. It is better to try different method of imputation and evaluate the consistency. Did you consider Race and overweight which are significantly correlated with loss to follow-up?

Data analysis:

Not sure what “multilevel multiple regression” is. It seems that it is better to use generalized linear mixed models (GLMM) for the clustered and repeated dichotomous measure. In the GLMM, the random cluster and time should be tested for significant. The time * intervention interaction should also be tested.

What baseline characteristics were adjusted in the model? How did you adjust for Race and overweight which are significantly correlated with loss to follow-up? This needs careful consideration and discussion.

If multiple children in the same family were recruited, was the correlation within the same family adjusted in your model?

For so many outcomes, p values need to be adjusted for multiple tests.

Results:

Table 1: separate overweight and obesity

Table 3 can be omitted. Add the p values to Table 4.

Similarly, for table 5 report OR, 95% CI and p values. Do not report regression coefficients.

Add p values to Table 6.

Needs more consideration and discussion about the discrepancies between ITT and per-protocol results.

6. PLOS authors have the option to publish the peer review history of their article (what does this mean?). If published, this will include your full peer review and any attached files.

Do you want your identity to be public for this peer review? For information about this choice, including consent withdrawal, please see our Privacy Policy.

Reviewer #1: No

---

## [Author Response · Author response to Decision Letter 0]

4 Feb 2022

Response to Editor and Statistical reviewer

PONE-D-21-30893

The effectiveness of a web-based Dutch parenting program to prevent overweight in children 9-13 years of age: results of a two-armed cluster randomized controlled trial

Response to comments Academic Editor PLOS ONE: Lee Van Horn, PhD

Dear Dr. Ruiter,

Thank you for submitting your manuscript to PLOS ONE. After careful consideration, we feel that it has merit but does not fully meet PLOS ONE’s publication criteria as it currently stands. Therefore, we invite you to submit a revised version of the manuscript that addresses the points raised during the review process.

The decision for this paper is a difficult one. Let me start by saying that I believe it is important to publish this paper, its just a difficult situation. On the one hand you have taken the appropriate steps, registering the clinical trial, publishing a study protocol, and then for the most part following that protocol in this paper. On the other hand, I have many of the same concerns that are raised by the statistical reviewer as well as a few others. I think you should know where I’m coming from as an editor, while I’m trained as a social scientist, my primary role is as a methodologist, I’ve been the primary statistical methods person on multiple randomized and group randomized trials, some targeting physical activity and weight loss. I am also a reviewer for major clinical trials grants in the area. If I had reviewed either the grant application or the study protocol paper I would have had reservations about the statistical methods proposed (doing analyses in Mplus with TYPE = COMPLEX, failure to account for multiple testing, dichotomizing outcomes, treating T1 and T2 as separate outcomes, reporting both ITT and completers only analysis) and would have requested major revisions. So, now the question I’m facing is how to reconcile the desire to stay true to the published protocol paper versus moving forward with results that do not meet the standard I would expect from a clinical trial. My overall decision at this point is to request major revisions which try to address these issues. I’ll also note that I held off sending this paper out to content specific reviewers because I wanted to see if the statistics reviewer had the same concerns I did, and because the substantive reviews will likely be better if these other issues are fixed.

Several things seem straight forward to me: 

1) the protocol paper did not include per-protocol analyses, I believe that these need to be dropped or if they are retained they need to be done using Complier Average Causal Effects analyses appropriate for cluster randomized trials (there is a paper out there on doing this by Booil Jo), but these are difficult analyses to do and I would want the code and output included so that they could be checked; 

- Thank you for your advice. For this reason we removed the per-protocol analyses. In the Method section in line 372-374 we removed the text “, and in a per-protocol framework (i.e., parent-child dyads who completed all three questionnaires and parents in the intervention group who completed at least two of the e-learning modules)”. 

- In the result section in line 513-525 and in line 564-574 we removed the results of the per-protocol analyses.

- In the discussion section in line 618-619 we removed the text “and per protocol analyzes”.

- In the abstract in line 41-45 and the discussion section in line 621-625 we removed the text “Per protocol analyses also revealed additional positive effects with respect to the child’s consumption of fruits and vegetables, playing outside, being physical active, and the parents’ use of an authoritative general parenting style, parental control over eating, monitoring of the child’s physical activity, and higher satisfaction regarding parental self-efficacy”.

2) the protocol paper did not mention how multiple outcomes would be tested, this is really a major problem, you have 13 outcomes multiplied by two different reporters (parents and children) and two different time points, so 52 different analyses for 3 different types of analyses. The trial registration includes 7 primary outcomes, only a little bit better. This needs to be addressed and I’m not sure that you have many options at this point, but the end result is likely that there are no reliable differences between treatment and control groups; 

- Thank you for this comment. We reduced our reported primary outcomes to 7, as mentioned in our protocol paper.

- The 13 outcomes we reported in our in our first submitted main manuscript were: 

1. Breakfast (days/week), daily Breakfast

2. Vegetables (days and amount/week), Daily vegetables

3. Daily 3 serving spoons of vegetables

4. Fruit (days and amount/week), Daily Fruit

5. Daily 2 pieces of fruit

6. SSB (days and glasses/week), Daily <2 glasses SSBs

7. Television (days and minutes/week)

8. Computer (days and minutes/week)

9. Screentime (days and minutes/week), Daily ≤120 min screen time

10. Play outside (days and minutes/week), Daily Play outside

11. Daily Play outside > 1 hour

12. Physical active (days and minutes/week), Dutch standard healthy exercise (participating in an organized sport for ≥30 minutes at least twice a week)

13. Walk/cycle to school (days/week)

- In our revised manuscript we now reported the following 7 primary outcomes as mentioned in the protocol paper:

1. Breakfast (days/week), daily Breakfast

2. Vegetables (days and amount/week), Daily vegetables

3. Fruit (days and amount/week), Daily 2 pieces of fruit

4. SSB (days and glasses/week), Daily <2 glasses SSBs

5. Screentime (days and minutes/week), Daily ≤120 min screen time

6. Play outside (days and minutes/week), Daily Play outside > 1 hour

7. Physical active (days and minutes/week), Dutch standard healthy exercise (participating in an organized sport for ≥30 minutes at least twice a week)

- Furthermore we now corrected for multiple outcome. P-values of interaction effects were corrected using Holm-Bonferroni correction. The p-values for continuous outcomes reported by children are corrected for the number of p-values for continuous outcomes reported by children. The p-values for continuous outcomes reported by parents are corrected for the number of p-values for continuous outcomes reported by parents. We did the same for the dichotomous variables. Pairwise comparisons were adjusted using Tukey-method to adjust for multiple testing. Accordingly, in the abstract in line 38 we added the text “after correction for multiple testing”, in the method section in line 408-409 we added the text: “To adjust for multiple testing, the p-values of the time x condition interaction effects were adjusted using Holm-Bonferroni sequential correction.” In the method section in line 418-419 we added the text: “For pairwise comparison, p-values were adjusted using the Tukey-method to adjust for multiple testing.”

- In the result section in line 488 and in line 537 we added the test “, and after adjusting for multiple testing,”

- In the Discussion section in line 709-721 we deleted the text: “First, it is worth noting that we did not apply Bonferroni correction for multiple testing, as this was the first study to examine the effects of an e-learning program for parents. In this exploratory study, we examined the effects of all relevant variables for which we expected the e-learning program to have an effect. Moreover, we did not expect the program to have particularly large effects, and we did not want to prematurely conclude that the e-learning program did not have an effect. Furthermore, we anticipated wide variations with respect to the outcome measures, as the e-learning program could have had affected slightly different variables among the children and parents (for example, the consumption of SSBs but not fruit could have been affected in one child, while the amount of screen time could have been affected in another child). Thus, applying Bonferroni correction may have been overly strict, as this conservative method can obscure significant results unless the effects and/or sample sizes are extremely large. Furthermore, this approach can increase the likelihood of a type II error, rendering a bona fide difference non-significant [69].”

- In addition, the following secondary outcomes were measured and reported in the revised manuscript:

1. parental styles

2. parenting practices

3. parental self-efficacy

4. parents’ willingness to follow the e-learning program

5. parents’ satisfaction with the e-learning program

3) Your actual statistical analyses need to be made clear. You simply report that you used Mplus with type=complex, this is not the statistical analysis, it is just the software used. As the statistical reviewer mentions, the models used need to be reported in careful detail with all predictors and outcomes specified, assumptions tested, and variance components reported; 

- Thank you for this comment. Based on your comments we decided to revise our strategy for analyses and to redo our analyses. We replaced the analyses in Mplus with linear mixed models in R. We now also more clearly describe how the analyses were conducted.

- The Statistical analyses section has been rewritten. In the method section in line 379-389 we deleted the text about the analysis in Mplus and replaced this text in line 374-379 by “To determine the effectiveness of the intervention, we used linear mixed models in R [66], instead of the planned regression analyses in Mplus as described in the study protocol [40]. Linear mixed models were chosen for two reasons: i) because of the nested data structure of the data (i.e., repeated measurements nested within participants, and participants nested in schools), and ii) because multi-level analyses do not delete participants listwise, and can more appropriately handle data missing at random [67, 68].”. And in line 389-419 “The multicategory variables for parental EBRB rules were dichotomized in order to compare between “no rules” and “indulgent/strict rules” and between “strict rules” and “no/indulgent rules”, providing better interpretation of data. Linear mixed models for continuous outcomes and generalized linear mixed models with logit links for binary outcomes were used. The package lme4 was used for both LMMs and GLMMs [69]. All models included a random effect for students to account for repeated measurements within participants. For each outcome, it was also tested whether it is necessary to additionally account for nesting of students within schools. This additional random factor was only included if the intraclass correlation for the school-level was higher than .05, suggesting significant clustering of students within schools [70, 71]. In addition, it was tested whether a random intercept and slope model significantly better fit the data than the more parsimonious random intercept model using log-likelihood ratio tests. If this was the case, the model with random slopes was used. If the more complex models including additional random factors and/or random slopes did not converge or were singular, we used a more parsimonious model, for example by only modelling random intercepts. Time, condition and their higher-order interactions were included as fixed effects in all models. An unstructured covariance structure was used for all models. Because there were two possible starting points of the study, namely i) February 2013, in which participants of 9 schools were randomized, and ii) September 2013, in which participants of another 2 schools were randomized, the variable ‘moment of randomization’ was included as covariate to correct for possible seasonal effects. To adjust for multiple testing, the p-values of the time x condition interaction effects were adjusted using Holm-Bonferroni sequential correction. All models were visually inspected for homogeneity and normal distribution of errors using residuals versus fitted value plots and Q-Q plots. If it was suspected that these assumptions might have been violated, or if a continuous outcome was strongly left or right skewed, we additionally fitted the models with beta-regression using the glmmTMB package [68]. The results from the beta-regression models were then compared with linear mixed models. Since the outcomes from the linear beta models did not lead to substantially different conclusions, only linear mixed models were reported for continuous outcomes. To determine treatment effects at each of the follow-up timepoints, pairwise post-hoc comparisons of estimated marginal means by time and condition were performed using the emmeans package. For paiwise comparison, p-values were adjusted using the Tukey-method to account for multiple testing. An α of 0.05 was used to indicate statistical significance for all analyses.”

4) the justification for the sample size calculations needs to be made in much more detail including both the ICC assumptions and the effect size assumptions along with references; 

- Thank you for this comment. We assume an intraclass correlation coefficient (ICC) of 0.05. This estimate is based on a study on ICCs for obesity indicators in primary schools (Gray et al., 2016). The ICC’s for most outcomes varied between 0.02 and 0.05. Based on this study we decided to use an ICC of 0.05 for the power analysis.

- As for the effect size assumption: The assumption of an increase of 20% is based on previous effectiveness studies and school-based programs in children 9-13 years of age [48-51]”. 

- 48. De Lijster-van Kampen GPA, Gomez-Tromp M, Kocken PL: Evaluatie Door Dik en Dun: Resultaten van een pilotonderzoek naar de effecten van een behandelprogramma bij kinderen met overgewicht of obesitas in de leeftijd van 8-12 jaar [Results of a pilot study on the effects of a treatment program for overweight or obese children aged 8-12 years]. In.: TNO Kwaliteit van Leven; 2010.

- 49. Howerton MW, Bell BS, Dodd KW, Berrigan D, Stolzenberg-Solomon R, Nebeling L: School-based nutrition programs produced a moderate increase in fruit and vegetable consumption: meta and pooling analyses from 7 studies. J Nutr Educ Behav 2007, 39(4):186-196.

- 50. Reynolds KD, Franklin FA, Binkley D, Raczynski JM, Harrington KF, Kirk KA, Person S: Increasing the fruit and vegetable consumption of fourth-graders: results from the high 5 project. Prev Med 2000, 30(4):309-319.

- 51. Robinson TN: Reducing children's television viewing to prevent obesity: a randomized controlled trial. Jama 1999, 282(16):1561-1567.

- We refer here to the text in our protocol paper: “Sample size was calculated based on the difference between the intervention group and the control group in terms of the following outcomes: improvement in healthy eating and a reduction in sedentary behavior in children in accordance with Dutch standards. Because the e-learning program has not been tested previously with respect to effectiveness, it is difficult to estimate an effect size. However, based on the outcomes of similar effectiveness studies and school-based programs in children 9-13 years of age [16-19], we expect a minimum difference of 20% between the control and intervention groups 12 months after baseline in the children’s dietary and/or sedentary behavior in accordance with the Dutch standards.”.

- In the method section in line 214-215 we deleted the text: “(with an estimated intraclass correlation coefficient of 0.05 and an average cluster size of 14).”. and in line 215-217 we added the text “We assume an average cluster size of 14 and an intraclass correlation coefficient (ICC) of 0.05 based on a study on ICCs for obesity indicators in primary schools in which the ICCs for most outcomes were between 0.02 and o.05 [47].”. And in line 221-223 we added the text “The assumption of an increase of 20% is based on previous effectiveness studies and school-based programs in children 9-13 years of age [48-51]”.

5) Missing data and imputation needs to be described in more detail.

- Thank you for this comment. We replaced our current analyses in Mplus with analyses with linear mixed models in R, that more appropriately handles missing data at random. We rewrote our Statistical analyses section to make this clearer. In the method section in line 379-389 we deleted the text about the analysis in Mplus and replaced this text in line 374-379 by “To determine the effectiveness of the intervention, we used linear mixed models in R [66], instead of the planned regression analyses in Mplus as described in the study protocol [40]. Linear mixed models were chosen for two reasons: i) because of the nested data structure of the data (i.e., repeated measurements nested within participants, and participants nested in schools), and ii) because multi-level analyses do not delete participants listwise, and can more appropriately handle data missing at random [67, 68].”

Now the difficult things. First of all is your use of Mplus and type=complex to account for the clustering in the data. I’m going to start by nothing that you are wrong about this being multilevel analysis, when you use the type=complex option Mplus simply adjusts the standard errors of your effects for clustering using a sandwich estimator, but this is not a multilevel regression model. The first reason I don’t typically suggest Mplus for multilevel models is that users often don’t understand what is happening, the second is that many Mplus features are not well documented and it really isn’t clear what sandwich estimator is employed. In any case, this should be similar to more typical implementations of GEE. While GEE can be acceptable in GRTs for correcting for clustering, it isn’t very efficient and the approach recommended by the reviewer is clearly the standard. A recent paper by Turner et al (American J of Public Health) discusses some of these issues. If the protocol paper wasn’t published I would feel quite strongly (along with the stats reviewer) that the current analyses should be completely replaced with the conventional multilevel models recommended by the reviewer. However, given the existence of the protocol paper, my suggestion is that the ITT and completers analyses stays as is (with the full model including all covariates and variance parameter estimates described) but that the analyses suggested by the reviewer be included in an appendix and any discrepancies noted. There really shouldn’t be large discrepancies if everything was done right, and my reason for asking for this is that it will greatly increase my confidence that everything was done right.

- Thank you for your comment and advice. Based on your comments we decided to revise our strategy for analyses and to use the multilevel models recommended by you and the statistical reviewer. As we reported as answer on you question 3, we have rewritten our Statistical analyses section, see in the Method section in line 374- 419 (see above), and we now report the results of these new analyses in the Results section in line 482-492 “The effects of the e-learning program on the children’s EBRBs and parenting dimensions according to the results of the intention-to-treat analysis are summarized in Tables 3 and 4, and the completers-only sensitivity analyses are shown in Additional file 4 (S4 Appendix) Effects of the e-learning program on the children’s EBRBs

- Based on both the children’s responses and the parents’ responses to the questionnaires, and after adjusting for multiple testing, we found no significant differences in the changes over time between the intervention and control condition with respect to the children’s diet, sedentary behavior or physical activity behavior. In addition, post-hoc pairwise comparisons showed no significant difference between conditions at any of the follow-up times.” and in line 533-541 “Effects of the e-learning program on parenting dimensions

- Based on both the children’s responses and the parents’ responses to the questionnaires, and after adjusting for multiple testing, we found no significant differences in the change over time between the intervention and control condition with respect to parenting dimensions, including general parenting, specific EBRB-guided parenting (e.g., monitoring, modeling, and establishing EBRB rules), or parental self-efficacy. In addition, post hoc pairwise comparisons showed no significant difference between conditions at any of the follow-up times.” and in line 577 Table 3 and Table 4 .

- We decided not to report our analyses in Mplus in an additional file. This decision was made because of the limitations of these analyses as pointed out by the reviewer and editor, and to make the manuscript more readable.

Second, were covariates included in these analyses? Also, was the analyses the outcome at time T controlling for baseline or was it change scores? If covariates were not included, why not? I agree with the reviewer that baseline values for weight as other variables should have been included, the protocol paper is not clear about how or if this will be done.

- In our Mplus analyses no covariates were included, because there were no differences at baseline between the variables. 

- The analyses in Mplus measured the change scores.

- In the revised manuscript we replaced the analyses in Mplus with linear mixed models in R. Time, condition and their higher-order interactions were included as fixed effects in all models. An unstructured covariance structure was used for all models. Because there were two possible starting points of the study (i) participants of 9 schools who were randomized in February 2013, ii) and participant of another 2 schools who were randomized in September 2013), the variable ‘moment of randomization’ was included as covariate to correct for possible seasonal effects. And in the method section in line 403-408 we added the text: “Time, condition and their higher-order interactions were included as fixed effects in all models. An unstructured covariance structure was used for all models. Because there were two possible starting points of the study, namely i) February 2013, in which participants of 9 schools were randomized, and ii) September 2013, in which participants of another 2 schools were randomized, the variable ‘moment of randomization’ was included as covariate to correct for possible seasonal effects.”

Third, I’m not sure that all of your effect sizes are even in the correct direction for the few significant treatment effects. Looks like there were more sweetened beverages consumed in the treatment than control groups? Also, some of the significance tests in the completers only table seem to be off? Oh, the use of superscripts to indicate which effects are significant for which set of analyses doesn’t work at all. You should probably just report the full results of the ITT and then, separately, of the completers-only analyses (again, dropping the protocol analyses).

- Thank you for this comment. As mentioned above, based on your comments we decided to revise our strategy for analyses and we replace our analyses with linear mixed models that account for the multilevel structure of the data as recommended by you and the statistical reviewer. 

- We replaced table 3-6 (with the original results of the M-plus analyses) with the results of our analyses in R. In our new tables (table 3 and 4) we show the results of our intention-to-treat analyses. In these tables we don’t use superscripts to indicate which effects were significant in the completers only analyses. The two tables with the results of our completers only analyses are placed in the additional file to make the manuscript easier to read.

- In the additional file S4 Appendix we report the full results of the completers-only analyses. 

I’ve noted several other issues. One is that I doubt that substantive reviewers would be impressed with your outcomes for the dietary or physical activity measures. On the trials I’ve been on we have ignored self or parent reported PA and focused exclusively on accelerometer measured PA because of concerns about bias as well as unreliability. Obviously you can’t change this now, but it needs to be a very important discussion point and the issues with the measures need to be made very clear.

- Thank you for this comment and advice. In our main manuscript in the Discussion section in line 727-735 we mentioned this study limitation: “Fourth, the information regarding the behaviors of the children and parents was obtained entirely using self-reporting questionnaires completed by the children and parents, which may have resulting in over-reporting and/or under-reporting due social desirability and/or recall bias. To minimize social desirability and optimize measurement validity, we stressed full confidentiality (i.e., anonymity) to our participants. To minimize recall bias, the interval between the events being recalled and the questionnaire was relatively short (participants were asked to recall events from the past month or week), which likely also increased the self-reporting reliability.” 

- Because of your advice in the Discussion section in line 730-731 we added the text “and may contribute to unreliability of the reported data [80].”, and in line 735-737 we added the text “ To increase reliability it may have been better to use accelerometers to report physical activity; however, for determining whether the children meet or did not meet the Dutch standard for healthy exercise, a questionnaire may be appropriate [85]. “ 

Another issue which is really major is that I do not think you have interpreted these results correctly. You conclude that, “our web-based parenting program yields promising effects…” and when I look at your results the ITT analyses (these should be the gold standard) truly show no effects if you do any meaningful correction for multiple comparisons. Again, it looks to me like some of your results are in the wrong direction which also suggests no overall effects. The completers only analyses looks about the same to me. My best guess looking at these results is that this study has more evidence of no effects than of effects, the only place where you see something is the per-protocol analyses but those results are well known to be biased the way you did them.

- Thank you for this comment and your advice. In the abstract section in line 48-52 we deleted the text “Our results show that the web-based parenting program has promising beneficial effects and can be easily incorporated into existing programs designed to prevent overweight in children.”. And in line 48-54 we added the our new conclusion “Although parents were generally satisfied with the parenting program, following this program had no significant beneficial effects regarding the children’s energy balance-related behaviors or the parenting skills compared to the control condition. This program may be more beneficial if used by high-risk groups (e.g. parents of children with unhealthy energy balance-related behaviors and/or overweight children) compared to the general population, warranting further study.”. And in the Discussion section in line 747-749 we added the text “Despite the parent’s overall satisfaction regarding the e-learning program and the ease with which our e-learning program can be integrated into the “Scoring for health” program, we found no significant differences between the intervention and control condition.”. In line 760-767 we changed the text into “Thus, the e-learning program may have beneficial effects if added to an existing overweight-prevention program currently being offered to high-risk groups, for example in the Youth Health Care or in a hospital setting, in which we expect the parents will feel a stronger sense of urgency and therefore be more motivated to use the e-learning program. Thus, future research should test the effectiveness of this e-learning program specifically in high risk groups, such as parents of children with unhealthy EBRBs, and/or overweight children, thus providing more information regarding the benefits of this program in this specific target population”.

My final note is that I disagree with the reviewer about dropping the tables with parameter estimates and standard errors, I would focus more rather than less on them.

- Thank you for this comment and your advice. We replaced table 3-6 because we completely rerun analyses in R. In our new tables (Table 3 and 4) for the continuous variables we presented the mean and standard deviation at T0, T1 and T2 and the p-values and adjusted p-values. For the dichotomous variables we presented the probabilities and standard deviation at T0, T1 and T2 and the p-values and adjusted p-values. 

NOTE: In our text we mixed the terms “group” and “condition”, and the terms “modules” and “episodes”. To make te text better readable in lines 155, 208, 209, 218, 219, 232, and 245, and we replaced the terms “control group” and “intervention group” into “control condition” and “intervention condition”, and in lines 230 and 231 we replaced the term “episodes” into “modules”.

Journal Requirements:

- We ensure that our manuscript meets PLOS ONE’s style requirements.

- In our published protocol paper the following text included “We asked the principals’ permission to distribute envelopes to the parents of the children in the participating schools. The envelopes contained an invitation letter in which we asked parents to participate with their child in our study. In addition, the envelopes contained information regarding the study (including the purpose of the study, length of the study, frequency of measurements, eligibility criteria, confidentially of the data, etc.), a passive informed consent form for the parent, a passive informed consent form for the child, and an envelope for returning the forms. Two weeks after distributing the invitation letters, we visited all of the school classes.”.

- In the Method section in line 173-176 we added the text “All parents and children received written information about the study [40]. In addition, one researcher (author ER) was present at all classes to provide verbal information regarding the study and answer any questions raised by the children. For this study, we used passive parental consent, which was approved by the Medical Review Ethics Committee.”.

3. Please provide the original, full-length study protocol, or project proposal, that you submitted to your ethics committee for review and approval prior to study commencement. This document should typically contain a background section, detailed description of the methods used (including information about dates of recruitment, participant characteristics, inclusion/exclusion criteria, detailed descriptions of the protocols/interventions utilized, plans for outcome assessment and data collection/analysis, and references). Please note that we require this to be submitted with your manuscript in both the original language version and a version translated into English.

- We uploaded our original project proposal in the Dutch language, called “METC_ELVO C1 Onderzoeksprotocol” and the English translation can be found in our published protocol named “The effectiveness of a web-based Dutch parenting program to prevent overweight in children 9–13 years of age: study protocol for a two-armed cluster randomized controlled trial”.

4. Thank you for submitting your clinical trial to PLOS ONE and for providing the name of the registry and the registration number. The information in the registry entry suggests that your trial was registered after patient recruitment began. PLOS ONE strongly encourages authors to register all trials before recruiting the first participant in a study.

a) your reasons for your delay in registering this study (after enrolment of participants started);

- We sent the questionnaires to parents after registration in the trial register and after permission from the medical ethics committee, but we have already approached school directors and the program manager of the existing school-based overweight prevention program “Scoring for Health” for possible participation. So that we could start immediately after approval. It was necessary to have their cooperation before we could submit the study design.

b) confirmation that all related trials are registered by stating: “The authors confirm that all ongoing and related trials for this drug/intervention are registered

- In the Method section in line 165-166 we added the following text “The authors confirm that all ongoing and related trials for this intervention are registered.”

"This study was funded by a grant from the Netherlands Organization for Health Research and Development (ZonMw; project number 505010296015), which played no role in the design of execution of this study, the analysis or interpretation of the data, or the decision to publish the results."

"This study was funded by a grant from the Netherlands Organization for Health Research and Development (ZonMw; project number 505010296015, 200100001). This funder had no role in study design, data collection and analysis, interpretation of the data, decision to publish the results, or preparation of the manuscript."

- No amended statement is required in our online Funding Statement.

- In the Declaration section in line 784-788-we deleted the following text: “Funding This study was funded by a grant from the Netherlands Organization for Health Research and Development (ZonMw; project number 505010296015, 200100001). This funder had no role in study design, data collection and analysis, interpretation of the data, decision to publish the results, or preparation of the manuscript.”

- In the Declaration section in line 791-794 we deleted the text: “This study was funded by a grant from the Netherlands Organization for Health Research and Development (ZonMw; project number 505010296015), which played no role in the design of execution of this study, the analysis or interpretation of the data, or the decision to publish the results.”.

- “No amended statement is required in our Funding Statement.”

6. We note that you have indicated that data from this study are available upon request. PLOS only allows data to be available upon request if there are legal or ethical restrictions on sharing data publicly. For information on unacceptable data access restrictions, please see http://journals.plos.org/plosone/s/data-availability#loc-unacceptable-data-access-restrictions. 

- There are no ethical or legal restrictions on sharing a de-identified data set.

- There are no ethical or legal restrictions on sharing a de-identified data set. We uploaded the minimal anonymized data set necessary to replicate our study findings to a repository of the Radboud University in the Netherlands, see <URL>.

- As soon as we have received the URL link we will forward it.

Reviewers' comments:

Reviewer's Responses to Questions

Comments to the Author

1. Is the manuscript technically sound, and do the data support the conclusions?

Reviewer #1: Partly

2. Has the statistical analysis been performed appropriately and rigorously?

Reviewer #1: No

- Based on the comments of the statistical reviewer and Editor, we revised our strategy for analyses and redid our analyses in the statistical package R. We described our revised strategy for analyses in the Statistical analyses section in line 374-419 , see above the full revised text in the answer to question 3 of the Academic Editor.

3. Have the authors made all data underlying the findings in their manuscript fully available?

Reviewer #1: Yes

4. Is the manuscript presented in an intelligible fashion and written in standard English?

Reviewer #1: Yes

5. Review Comments to the Author

Reviewer #1: Sample size: why the intra class correlation is set at a low level of 0.05?

- Thank you for this comment. We assume an intraclass correlation coefficient (ICC) of 0.05. This estimate is based on a study on ICCs for obesity indicators in primary schools (Gray et al., 2016). The ICC’s for most outcomes varied between 0.02 and 0.05. Based on this study we decided to use an ICC of 0.05 for the power analysis.

- In the method section in line 214-215 we deleted the text: “(with an estimated intraclass correlation coefficient of 0.05 and an average cluster size of 14).”. and in line 215-217 we added the text “We assume an average cluster size of 14 and an intraclass correlation coefficient (ICC) of 0.05 based on a study on ICCs for obesity indicators in primary schools in which the ICCs for most outcomes were between 0.02 and o.05 [47].”.

Missing data: what method was used for imputation? There are discrepancies between ITT and per-protocol analysis. It is better to try different method of imputation and evaluate the consistency. Did you consider Race and overweight which are significantly correlated with loss to follow-up?

- Thank you for this question. We replaced our current analyses in Mplus with analyses with linear mixed models (as recommended by you and the Academic Editor) in R, that more appropriately handles missing data at random. We rewrote our Statistical analyses section to make this clearer. In the method section in line 379-389 we deleted the text about the analysis in Mplus and replaced this text in line 374-379 by “To determine the effectiveness of the intervention, we used linear mixed models in R [66], instead of the planned regression analyses in Mplus as described in the study protocol [40]. Linear mixed models were chosen for two reasons: i) because of the nested data structure of the data (i.e., repeated measurements nested within participants, and participants nested in schools), and ii) because multi-level analyses do not delete participants listwise, and can more appropriately handle data missing at random [67, 68].”

Data analysis:

Not sure what “multilevel multiple regression” is. It seems that it is better to use generalized linear mixed models (GLMM) for the clustered and repeated dichotomous measure. In the GLMM, the random cluster and time should be tested for significant. The time * intervention interaction should also be tested.

- Thank you for this comment. We replaced our current analyses in Mplus type=complex with the multilevel models recommended by you and the Academic Editor. For this reasons we rewrote our Statistical analyses section. In the method section in line 379-389 we deleted the text about the analysis in Mplus and replaced this text by: “Linear mixed models for continuous outcomes and generalized linear mixed models with logit links for binary outcomes were used. The package lme4 was used for both LMMs and GLMMs [69]. All models included a random effect for students to account for repeated measurements within participants. For each outcome, it was also tested whether it is necessary to additionally account for nesting of students within schools. This additional random factor was only included if the intraclass correlation for the school-level was higher than .05, suggesting significant clustering of students within schools [70, 71]. In addition, it was tested whether a random intercept and slope model significantly better fit the data than the more parsimonious random intercept model using log-likelihood ratio tests. If this was the case, the model with random slopes was used. If the more complex models including additional random factors and/or random slopes did not converge or were singular, we used a more parsimonious model, for example by only modelling random intercepts. Time, condition and their higher-order interactions were included as fixed effects in all models. An unstructured covariance structure was used for all models. Because there were two possible starting points of the study, namely i) February 2013, in which participants of 9 schools were randomized, and ii) September 2013, in which participants of another 2 schools were randomized, the variable ‘moment of randomization’ was included as covariate to correct for possible seasonal effects. To adjust for multiple testing, the p-values of the time x condition interaction effects were adjusted using Holm-Bonferroni sequential correction. All models were visually inspected for homogeneity and normal distribution of errors using residuals versus fitted value plots and Q-Q plots. If it was suspected that these assumptions might have been violated, or if a continuous outcome was strongly left or right skewed, we additionally fitted the models with beta-regression using the glmmTMB package [68]. The results from the beta-regression models were then compared with linear mixed models. Since the outcomes from the linear beta models did not lead to substantially different conclusions, only linear mixed models were reported for continuous outcomes. To determine treatment effects at each of the follow-up timepoints, pairwise post-hoc comparisons of estimated marginal means by time and condition were performed using the emmeans package. For paiwise comparison, p-values were adjusted using the Tukey-method to account for multiple testing. An α of 0.05 was used to indicate statistical significance for all analyses.”

What baseline characteristics were adjusted in the model? How did you adjust for Race and overweight which are significantly correlated with loss to follow-up? This needs careful consideration and discussion.

- Thank you for these questions. No baseline characteristics were adjusted in the model. We found no difference between the intervention and control conditions at baseline regarding Race (Ethnicity) and overweight. As reported in our protocol paper, for that reasons we didn’t adjust for these characteristics. However, if desired, we could also additionally check whether including these covariates substantially changes the conclusions drawn from the models.

If multiple children in the same family were recruited, was the correlation within the same family adjusted in your model?

- Thank you for this question. See in the method section in line 207-209: “If more than one child living in the same household participated in the study, all participating children in that household were assigned to the same condition as the oldest participating child in order to avoid contamination between conditions.”.

For so many outcomes, p values need to be adjusted for multiple tests.

- Thank you for this comment. Based on your comment and the same comment of the academic editor, we now corrected for multiple testing. P-values of interaction effects were corrected using Holm-Bonferroni correction. Post-hoc pairwise comparisons were adjusted using Tukey-method to adjust for multiple testing. Accordingly, in the method section in line 408-409 we added the text: “To adjust for multiple testing, the p-values of the time x condition interaction effects were adjusted using Holm-Bonferroni sequential correction.” In the method section in line 418-419 we added the text: “For pairwise comparison, p-values were adjusted using the Tukey-method to adjust for multiple testing.”

- In the Discussion section in line 709-721 we deleted the text: “First, it is worth noting that we did not apply Bonferroni correction for multiple testing, as this was the first study to examine the effects of an e-learning program for parents. In this exploratory study, we examined the effects of all relevant variables for which we expected the e-learning program to have an effect. Moreover, we did not expect the program to have particularly large effects, and we did not want to prematurely conclude that the e-learning program did not have an effect. Furthermore, we anticipated wide variations with respect to the outcome measures, as the e-learning program could have had affected slightly different variables among the children and parents (for example, the consumption of SSBs but not fruit could have been affected in one child, while the amount of screen time could have been affected in another child). Thus, applying Bonferroni correction may have been overly strict, as this conservative method can obscure significant results unless the effects and/or sample sizes are extremely large. Furthermore, this approach can increase the likelihood of a type II error, rendering a bona fide difference non-significant [69].”

Results:

Table 1: separate overweight and obesity

- Thank you for this suggestion. In the Result section in line 451 in table 1 we now separate overweight and obesity.

Table 3 can be omitted. 

- Thank you for this suggestion. We omitted table 3 in the main manuscript. Because we replaced our current analyses in Mplus type=complex with linear mixed models recommended by you and the Academic Editor.

Add the p values to Table 4.

- Thank you for this suggestion. Since we reran all analyses we now report different statistics in Table 3 and 4. In our new tables (Table 3 and 4) we now present the p-values and adjusted p-values for interaction effects for the continuous variables as well as for the dichotomous variables.

Similarly, for table 5 report OR, 95% CI and p values. Do not report regression coefficients.

- Thank you for this suggestion. In our new tables (Table 3 and 4) for continuous variables we now present the means and standard errors at T0, T1 and T2 and the p-values and adjusted p-values for the interaction effects from linear mixed models. For the dichotomous variables we now present the observed probabilities and standard deviation at T0, T1 and T2 and the p-values and adjusted p-values for the interaction effects from generalized linear mixed models.

Add p values to Table 6.

- Thank you for this suggestion. As explained above, In our new tables (Table 3 and 4) we presented the p-values and adjusted p-values for the continuous variables as well as for the dichotomous variables.

Needs more consideration and discussion about the discrepancies between ITT and per-protocol results.

- Thank you for this comment. On advice of the Academic Editor we deleted the per-protocol results, because these were not described in our study protocol.

---

## [Decision Letter · Decision Letter 1]

9 Jun 2022

PONE-D-21-30893R1The effectiveness of a web-based Dutch parenting program to prevent overweight in children 9-13 years of age: results of a two-armed cluster randomized controlled trialPLOS ONE

Dear Dr. Ruiter,

Thank you for resubmitting your manuscript to PLOS ONE. After careful consideration, we feel that it has merit but does not fully meet PLOS ONE’s publication criteria as it currently stands. Therefore, we invite you to submit a revised version of the manuscript that addresses the points raised during the review process.

I have now obtained a second review from the original stats reviewer, a new review from an expert in this content area, and have reviewed the manuscript again myself. The new reviewer has raised a number of issues which should be addressed. My impression is that these can be done fairly quickly and I will do my best to make a decision quickly in the next round based on your response to these reviews.

We look forward to receiving your revised manuscript.

Kind regards,

Lee Van Horn, PhD

Academic Editor

PLOS ONE

Reviewers' comments:

Reviewer's Responses to Questions

**Comments to the Author**

1. If the authors have adequately addressed your comments raised in a previous round of review and you feel that this manuscript is now acceptable for publication, you may indicate that here to bypass the “Comments to the Author” section, enter your conflict of interest statement in the “Confidential to Editor” section, and submit your "Accept" recommendation.

Reviewer #1: All comments have been addressed

Reviewer #2: (No Response)

2. Is the manuscript technically sound, and do the data support the conclusions?

Reviewer #1: (No Response)

Reviewer #2: Yes

3. Has the statistical analysis been performed appropriately and rigorously? 

Reviewer #1: (No Response)

Reviewer #2: Yes

4. Have the authors made all data underlying the findings in their manuscript fully available?

Reviewer #1: (No Response)

Reviewer #2: (No Response)

5. Is the manuscript presented in an intelligible fashion and written in standard English?

Reviewer #1: (No Response)

Reviewer #2: Yes

6. Review Comments to the Author

Reviewer #1: (No Response)

Reviewer #2: Thank you for the opportunity to review a revised version of the manuscript, “The effectiveness of a web-based Dutch parenting program to prevent overweight in children 9-13 years of age: results of a two-armed cluster randomized controlled trial.” This paper presents the results of a cluster-randomized trial conducted in the Netherlands of a web-based parenting program known as “Making a healthy deal with your child.” The intervention consisted of five 30-minute modules that walked parents through challenging parenting situations specific to youth’s healthy lifestyle behaviors (e.g., struggles with child consuming vegetables, having excess screen time). While involving parents in childhood obesity-related interventions is the gold standard for treatment, school-based approaches are more common for prevention, with parent involvement and participation generally being more variable in these interventions. In this study, authors ask an important research question in that they aim to assess the value of adding an online self-guided parenting component to an existing school-based intervention, “Scoring for Health.” From my perspective, the authors were quite responsive to the comments of the Editor and statistical reviewer; I defer to them on the need for authors to make further analysis-related revisions.

The authors report on a large clinical trial that was registered and for which a protocol paper has been previously published. Further, authors describe the theoretical underpinnings of the program as coming from strong evidence-based positive parenting programs – parent effectiveness training and the Parent Management Training – Oregon Model. The authors report that they developed the intervention with qualitative input from mothers who lived in low-SES neighborhoods. The study team collected data at three time points, and participant retention rates were good, with 72% completing the 12-month follow up assessment. Authors also reported process evaluation data related to parents’ use of the intervention (e.g., log-in activity, module completion). My concerns include the use of a within-school rather than between-school design due to contamination effects as well as the exclusive use of self-reported measures to assess study outcomes (though the authors do mention both of these in their limitations section). Overall, I think this manuscript will add to the ongoing conversation in the literature about ways to involve parents in obesity prevention efforts using digital approaches. My specific recommendations regarding each section of the paper are detailed below. I hope the authors find these comments useful as they continue to move this work forward.

Abstract:

• Minor - The abstract entered in the manuscript submission portal does not match the abstract in the revised version of the manuscript.

Introduction:

• I encourage the authors to use people-first language when referring to children with overweight or obesity in the introduction and throughout the manuscript. Information and examples of people-first language specific to obesity may be found here: https://www.obesityaction.org/action-through-advocacy/weight-bias/people-first-language/ For example, rather than using the term “obese children,” use “children with obesity.”

• Authors may want to cite some more up-to-date review papers when discussing the role of parents in obesity prevention and treatment more broadly:

o St. George et al. (2020). A developmental cascade perspective of paediatric obesity: A systematic review of preventive interventions from infancy through late adolescence. Obesity Reviews, 21(2), e12939.

o Tomayko et al. (2021). Parent involvement in diet or physical activity interventions to treat or prevent childhood obesity: An umbrella review. Nutrients, 13(9), 3227.

And importantly, include a bit more in the way of parent involvement in digital lifestyle interventions:

o Hammersley ML, Jones RA, Okely AD. Parent-focused childhood and adolescent overweight and obesity eHealth interventions: a systematic review and meta-analysis. Journal of Medical Internet Research. 2016;18(7):e5893.

o Kemp BJ, Thompson DR, Watson CJ, McGuigan K, Woodside JV, Ski CF. Effectiveness of family-based eHealth interventions in cardiovascular disease risk reduction: A systematic review. Preventive Medicine. 2021:106608.

• The authors do a nice job distinguising general positive parenting practices from domain-specific positive parenting practices and their relevance to childhood obesity (there is helpful discussion on the distinction between these in a paper by Power, Sleddens, and colleagues (2013) published in Childhood Obesity – Contemporary research on parenting: Conceptual, methodological, and translational issues). However, it is not clear from the hypotheses on page 7 what specific EBRB parenting practices they will examine as secondary outcomes. Please list them here.

Minor:

• Remove the word “becoming” from the first sentence of the paper. Overweight and obesity in children is a major health concern.

• Missing the word “time” in the following sentence on page 6: “v) children spending an excessive amount of [time] using the computer and not wanting …”

Methods and Results: These sections seem to have greatly benefitted from responding to the suggestions of the Editor and statistical reviewer. I have only a few questions/comments:

1. It appears as though parents assigned to the intervention condition received an email up front with their log-in code/access to the modules and then a congratulatory email following the completion of each module. I was less clear as to whether there were any automated reminder emails or any follow-up from study staff encouraging parents to complete the modules --- this information is included in the discussion but might be helpful to note in the methods section.

2. Authors currently present process data as parents who completed one module or 2 or more modules. How many parents completed 3, 4, and 5 modules? I was interested in more granular details regarding module completion as I believe this may have important implications for understanding study findings.

3. It’s interesting that measures of height were taken using a tape measure rather than a stadiometer. Why was this instrument selected to assess height (maybe ease of transporting)?

4. Where the response options for the parental EBRB rules measure (i.e., strict, indulgent, no rules) part of the original measure or did the authors develop these for the current study? I wondered why an option around the presence of rules that were reasonable (somewhere in between strict and indulgent) was not included.

Discussion:

• Authors note “our findings indicate that the program has room for improvement, as the intervention was not used intensively enough.” I think this point may be understated and that authors could discuss the issue of dose received in more detail earlier in the discussion rather than waiting until the strengths/limitations section. It is difficult to ascertain whether or not the intervention ‘worked’ when uptake was low.

• Related to my comment above, engagement in eHealth interventions is arguably one of the biggest challenges in the field. To what extent did authors include their target population in the process of developing their intervention modules? Did authors follow up with families to understand reasons they did or did not view the modules? They speculate that participants may not have been the appropriate targets and that a higher risk group may have been more appropriate, which is a fair point, but other strategies for increasing engagement beyond email reminders may have been helpful. For example, Mohr and colleagues have argued that supportive accountability via human contact may be required to get participants to engage with digital interventions. See: Mohr et al. (2011). Supportive accountability: a model for providing human support to enhance adherence to eHealth interventions. Journal of Medical Internet Research, 13(1), e30)

7. PLOS authors have the option to publish the peer review history of their article (what does this mean?). If published, this will include your full peer review and any attached files.

Reviewer #1: No

Reviewer #2: No

---

## [Author Response · Author response to Decision Letter 1]

27 Jul 2022

Response to Reviewers

Response to Reviewer #1

Thank you very much for your comment that all comments in our previous revised version have been addressed.

Response to Reviewer #2

Review #2 Comments to the Author

Thank you for the opportunity to review a revised version of the manuscript, “The effectiveness of a web-based Dutch parenting program to prevent overweight in children 9-13 years of age: results of a two-armed cluster randomized controlled trial.” This paper presents the results of a cluster-randomized trial conducted in the Netherlands of a web-based parenting program known as “Making a healthy deal with your child.” The intervention consisted of five 30-minute modules that walked parents through challenging parenting situations specific to youth’s healthy lifestyle behaviors (e.g., struggles with child consuming vegetables, having excess screen time). While involving parents in childhood obesity-related interventions is the gold standard for treatment, school-based approaches are more common for prevention, with parent involvement and participation generally being more variable in these interventions. In this study, authors ask an important research question in that they aim to assess the value of adding an online self-guided parenting component to an existing school-based intervention, “Scoring for Health.” From my perspective, the authors were quite responsive to the comments of the Editor and statistical reviewer; I defer to them on the need for authors to make further analysis-related revisions.

The authors report on a large clinical trial that was registered and for which a protocol paper has been previously published. Further, authors describe the theoretical underpinnings of the program as coming from strong evidence-based positive parenting programs – parent effectiveness training and the Parent Management Training – Oregon Model. The authors report that they developed the intervention with qualitative input from mothers who lived in low-SES neighborhoods. The study team collected data at three time points, and participant retention rates were good, with 72% completing the 12-month follow up assessment. Authors also reported process evaluation data related to parents’ use of the intervention (e.g., log-in activity, module completion). My concerns include the use of a within-school rather than between-school design due to contamination effects as well as the exclusive use of self-reported measures to assess study outcomes (though the authors do mention both of these in their limitations section). Overall, I think this manuscript will add to the ongoing conversation in the literature about ways to involve parents in obesity prevention efforts using digital approaches. My specific recommendations regarding each section of the paper are detailed below. I hope the authors find these comments useful as they continue to move this work forward.

Abstract:

• Minor - The abstract entered in the manuscript submission portal does not match the abstract in the revised version of the manuscript.

Thank you for your attentiveness. When submitting the revised version, I will ensure that the correct abstract text is stated.

Introduction:

• I encourage the authors to use people-first language when referring to children with overweight or obesity in the introduction and throughout the manuscript. Information and examples of people-first language specific to obesity may be found here: https://www.obesityaction.org/action-through-advocacy/weight-bias/people-first-language/ For example, rather than using the term “obese children,” use “children with obesity.”

Thank you very much for sharing this knowledge. We would like to contribute to the use of People-first language. For that reason, we have changed “overweight child” into “child with overweight” throughout the manuscript.

In the abstract section in line 46 we changed “overweight children” into “children with overweight”.

In the introduction section in line 58 we changed “overweight and obese children” into “children with overweight and obesity”. In line 59-61 we changed “In addition, families with low socio-economic status (SES) and families of Turkish and Moroccan descent have a particularly high prevalence of overweight children in the Netherlands” into “In addition, in the Netherlands families with low socio-economic status (SES) and families of Turkish and Moroccan descent have a particularly high prevalence of children with overweight”.

In the discussion section in line 547 we changed “overweight and/or obese children” into “children with overweight and/or obesity”. In line 550 and line 668 we changed “overweight children” into “children with overweight”.

• Authors may want to cite some more up-to-date review papers when discussing the role of parents in obesity prevention and treatment more broadly:

o St. George et al. (2020). A developmental cascade perspective of paediatric obesity: A systematic review of preventive interventions from infancy through late adolescence. Obesity Reviews, 21(2), e12939.

o Tomayko et al. (2021). Parent involvement in diet or physical activity interventions to treat or prevent childhood obesity: An umbrella review. Nutrients, 13(9), 3227.

And importantly, include a bit more in the way of parent involvement in digital lifestyle interventions:

o Hammersley ML, Jones RA, Okely AD. Parent-focused childhood and adolescent overweight and obesity eHealth interventions: a systematic review and meta-analysis. Journal of Medical Internet Research. 2016;18(7):e5893.

o Kemp BJ, Thompson DR, Watson CJ, McGuigan K, Woodside JV, Ski CF. Effectiveness of family-based eHealth interventions in cardiovascular disease risk reduction: A systematic review. Preventive Medicine. 2021:106608.

Thank you very much for the more up-to-date review papers, this really helps in discussing the role of parents in obesity prevention and treatment more broadly.

We inserted the reference of the review of St. George et al. (2020) in the introduction section in line 74 and 80, and the review of Tomayko et al. in line 77. In the discussion section in line 551-555 we added the text “However, systematic reviews by both Bleich et al. and Wang et al. showed that the most promising interventions are delivered in school settings that include a home component [82, 83]. In addition, based on their review, Tomayko et al.’s findings support including a parental component in school-based treatment and prevention programs designed to improve the child’s weight and/or weight status outcome [23].”.

23 Tomayko, E.J., et al., Parent involvement in diet or physical activity interventions to treat or prevent childhood obesity: An umbrella review. Nutrients, 2021. 13(9): p. 3227.

82. Bleich, S.N., et al., Interventions to prevent global childhood overweight and obesity: a systematic review. The Lancet Diabetes & Endocrinology, 2018. 6(4): p. 332-346.

83. Wang, Y., et al., What childhood obesity prevention programmes work? A systematic review and meta‐analysis. Obesity reviews, 2015. 16(7): p. 547-565.

We inserted the reference you suggest, namely the systematic review of Hammersley et al, and Kemp et al. in the introduction section in line 101. And in line 99-101 we added the text “Moreover, systematic reviews have shown that family-based eHealth interventions may be promising as part of a primary prevention-based strategy designed to improve children’s health [39, 40].”.

• The authors do a nice job distinguising general positive parenting practices from domain-specific positive parenting practices and their relevance to childhood obesity (there is helpful discussion on the distinction between these in a paper by Power, Sleddens, and colleagues (2013) published in Childhood Obesity – Contemporary research on parenting: Conceptual, methodological, and translational issues). However, it is not clear from the hypotheses on page 7 what specific EBRB parenting practices they will examine as secondary outcomes. Please list them here.

Thank you for your advice. To clarify the hypothesis, we have added what specific EBRB parenting practices we will examine as secondary outcomes. In the introduction section in line 151-152 we added the text “(monitoring, modeling, and applying rules regarding EBRBs)”. The text in line 150-152 now reads as “The secondary objectives of the e-learning program include strengthening parenting styles, improving specific parenting practices (monitoring, modeling, and applying rules regarding EBRBs), and increasing parental self-efficacy.”.

Minor:

• Remove the word “becoming” from the first sentence of the paper. Overweight and obesity in children is a major health concern.

Thank you for this comment. As you suggested in the introduction section in line 58 we removed the word “becoming”.

• Missing the word “time” in the following sentence on page 6: “v) children spending an excessive amount of [time] using the computer and not wanting …”

Thank you for your attentiveness. As you suggested in the introduction section in line 132 we added the word “time”.

Methods and Results: These sections seem to have greatly benefitted from responding to the suggestions of the Editor and statistical reviewer. I have only a few questions/comments:

1. It appears as though parents assigned to the intervention condition received an email up front with their log-in code/access to the modules and then a congratulatory email following the completion of each module. I was less clear as to whether there were any automated reminder emails or any follow-up from study staff encouraging parents to complete the modules --- this information is included in the discussion but might be helpful to note in the methods section.

Thank you for this comment. All parents who were allocated to the intervention condition received an email with their log-in code/access to the modules and then a congratulatory email following the completion of each module in this email the parents was also encouraged to complete the next module. In addition, the study staff send two reminder e-mails to the parents asking them to complete the e-learning program.

For clarification and to make the text more readable, in the methods section 228-229 we replaced the text “The parents in the control condition received only the brochure.” to line 222-223. Next, in line 224 we added the text “in the intervention condition”. Last, we have added the following text in line 226-228 “In addition, we sent reminder e-mails 7 and 12 weeks after providing the log-in code, reminding parents to complete the e-learning program.”.

2. Authors currently present process data as parents who completed one module or 2 or more modules. How many parents completed 3, 4, and 5 modules? I was interested in more granular details regarding module completion as I believe this may have important implications for understanding study findings.

Thank you for this question.

As mentioned in the result section in line 496-501: In the intention-to-treat framework, 171 (68%) of the 252 parent-child dyads in the intervention condition actually started the e-learning program; 109 (64%) of these parents completed 2 or more modules, and 85 parents (50 %) completed all 5 modules in the e-learning program (dose received). In addition, 81 (32%) parent-child dyads of the 252 parent-child dyads in the intervention condition did not start the e-learning program.

Below you can see how many parents completed 1, 2, 3, 4, and 5 modules of the total of 252 parent-child dyads in the intervention condition. The percentage of number of parents in the table below are the percentages of the total of 252 parent-child dyads in the intervention group.

It is important to know that the group of parent-child dyads which completed 0 episodes (n=122), consist of i) parents who did not start the e-learning (n=81), and ii) parents who start the e-learning but did not finish any episode (n=41).

Number of episodes completed

(N) Number of parents of the intervention group

(%)

0 episodes 122 (48.4)

1 episode or more 130 (51.6)

2 episodes or more 109 (43.3)

3 episodes ore more 92 (36.5)

4 episodes or more 88 (34.9)

5 episodes or more 85 (33.7)

In the result section in line 497 we replaced (68%) to line 496 to make the text more readable. Additionally, in line 496 we replaced “framework” into “approach”, in line 497 we added “109” and “171” and in line 498 we added “64%” and the text “and 85 parents (50%) completed all 5 modules”. The text in line 496-499 now reads as: “In the intention-to-treat approach, 171 of the 252 parent-child dyads (68%) in the intervention condition started the e-learning program; 109 of these 171 parents (64%) completed 2 or more modules, while 85 of these parents (50%) completed all 5 modules of the e-learning program.”

In Table 5 in S5 Appendix we added an extra column “completed all 5 modules”. The appendix is now named as “S5 Appendix_revised”.

3. It’s interesting that measures of height were taken using a tape measure rather than a stadiometer. Why was this instrument selected to assess height (maybe ease of transporting)?

A tape measure was used because this instrument was at that time used by the Youth health care department of the community health services, also because of the ease of transporting.

4. Where the response options for the parental EBRB rules measure (i.e., strict, indulgent, no rules) part of the original measure or did the authors develop these for the current study? I wondered why an option around the presence of rules that were reasonable (somewhere in between strict and indulgent) was not included.

Thank you for this question. The 3 response options for parental EBRB rules (strict, indulgent, no rules) were taken exactly from the original measure. The question about parental EBRB rules is derived from the Child Monitor questionnaire. This Child Monitor questionnaire is part of the "Local and National Health Monitor" in the Netherlands, which uses standard questions. The Child Monitor is part of a monitoring cycle and is required by the Dutch Public Health Act. Each four years a cross-sectional survey is conducted to gain insight into the population’s health, lifestyle and well-being of children age 6 months to 12 years [1] [2, 3]. The questions in the Child Monitor questionnaire about establishing EBRBs rules were only for parents of children 4-12 years of age.

1. Van der Star, M., Kindermonitor 2010: Gezondheidsonderzoek kinderen 0-12 jaar regio Nijmegen [Child Monitor 2010: Health research in children 0-12 years of age in the Nijmegen region, the Netherlands]. 2010, GGD Nijmegen: Nijmegen.

2. Van der Star, M., Regiorapport Kindermonitor 2009/2010: Gezondheid, welzijn en leefwijze van 0-12 jarigen in de regio Nijmegen. [Region report Child Monitor 2009/2010: Health, well-being and lifestyle of children 0 to 12 years of age in the Nijmegen region]. 2010, GGD Gelderland-Zuid: Nijmegen.

3. Ruiter, E., et al., Parents' underestimation of their child's weight status. Moderating factors and change over time: A cross-sectional study. PloS one, 2020. 15(1): p. e0227761.

To clarify, in the method section in line 321-324 we added the text “Parental EBRB rules were assessed via the parental and child questionnaire. Seven items were derived from the Child Monitor Questionnaire, which is part of the "Local and National Health Monitor" in the Netherlands [57]. Children and parents were asked…”. In line 330-332 we added text with the brief three definitions of the three categories. The text now reads as: “The children and parents were instructed to answer each question with the following possible answers: ‘i) yes, and we stick to them (“strict rules”); ii) yes, but we are flexible with them (“indulgent rules”), or iii) no, we have no rules about it (“no rules”).”.

Discussion:

• Authors note “our findings indicate that the program has room for improvement, as the intervention was not used intensively enough.” I think this point may be understated and that authors could discuss the issue of dose received in more detail earlier in the discussion rather than waiting until the strengths/limitations section. It is difficult to ascertain whether or not the intervention ‘worked’ when uptake was low.

Thank you for this comment and your advice. We discussed “our findings indicate that the program has room for improvement, as the intervention was not used intensively enough.” before the strengths and limitation section, namely in line number 555-556. However, we agree that we can discuss the issue of dose received in more detail as you advised. In the method section in line 341-342 we added the text “(dose received) of the in total 5 modules delivered in the e-learning program (dose delivery).”. The text in line 340-342 now reads as “This approach also provided information regarding the number of modules each parent completed (dose received) of the in total 5 modules delivered in the e-learning program (dose delivery).”. In the result section in line 499 we added the text “(dose received)”, the text in line 496-499 now reads as “In the intention-to-treat approach, 171 of the 252 parent-child dyads (68%) in the intervention condition actually started the e-learning program; 109 of these 171 parents (64%) completed 2 or more modules, and 85 of these parents (50 %) completed all 5 modules of the e-learning program (dose received).”. In the discussion section in line 556-557 we changed the text “The intensity of the program is also a serious point that warrant discussion.” into “The intensity at which the parents in our study were exposed to the intervention also warrants discussion.”. Next, in line 557-569 we added the text “We found that 64% of parents in the intervention condition who started the e-learning program completed 2 or more modules, with 50% of parents completing all 5 modules (the total delivery dose). However, if we consider all 252 parent-child dyads in the intervention condition (i.e., both the group of parents who started the program and the group of parents who never started the program), only 43% and 34% of all parents assigned to the intervention condition completed 2 or more modules or all 5 modules, respectively. Following 2 or more of the 5 modules is the minimum requirement to be considered sufficiently exposed to the e-learning program. Thus, in our study only 43 % of the total intervention condition was sufficiently exposed to the intervention. Therefore, the relatively low percentage of parents who were sufficiently exposed to the intervention may explain why we found no apparent difference between the intervention and control conditions, making it difficult to determine whether the intervention was actually effective given the high percentage of parents (57%) who were not sufficiently exposed to the intervention.”.

Last, in line 575 - 576 we deleted the text “For example, if parents do not feel a sense of urgency to focus on healthy EBRBs, it may not be self-evident to complete all modules.”, and we replaced the text “On the other hand, the amount of time parents needed to complete the program is relatively short, with 5 modules lasting approximately 30 minute each; thus, program was not necessarily expected to have a robust effect.” from line 576-578 to line 645-647.

• Related to my comment above, engagement in eHealth interventions is arguably one of the biggest challenges in the field. 

To what extent did authors include their target population in the process of developing their intervention modules? 

Thank you for this question. We include our target population in the process of developing our intervention modules by organizing focus group conversations. The aims of this focus group study were to i) explore and identify everyday life situations in which mothers experience difficulties stimulating healthy EBRBs in their school-age child, and ii) identify the reasons why mothers encounter these difficulties. The e-learning program’s content was based on the results of these focus groups [1]. Before testing the e-learning in an RCT, the e-learning was viewed by 5 mothers of different ethnic backgrounds who participated in the focus group study. The e-learning was optimized with the feedback obtained from these mothers.

1. Ruiter, E.L., et al., Everyday life situations in which mothers experience difficulty stimulating healthy energy balance–related behavior in their school-age children: a focus group study. BMC public health, 2019. 19(1): p. 701.

Did authors follow up with families to understand reasons they did or did not view the modules? 

Unfortunately, it did not fit within our research project to follow up with families to understand reasons they did or did not view the modules. This would have been meaningful and could be a recommendation for further research.

They speculate that participants may not have been the appropriate targets and that a higher risk group may have been more appropriate, which is a fair point, but other strategies for increasing engagement beyond email reminders may have been helpful. For example, Mohr and colleagues have argued that supportive accountability via human contact may be required to get participants to engage with digital interventions. See: Mohr et al. (2011). Supportive accountability: a model for providing human support to enhance adherence to eHealth interventions. Journal of Medical Internet Research, 13(1), e30)

Thank you for this comment. In the discussion section in line 578-603 we added the text “In addition, we used a relatively minimal engagement strategy. Our goal was to develop a self-guided web-based parenting program that can be added easily to existing overweight-prevention programs such as existing school-based/population-based interventions, thereby maximizing integration and minimizing costs. In one respect, this minimal-engagement strategy was successful, as 68% of parents started the intervention, with 64% of them completing at least 2 of the 5 modules, and 50% completing all 5 modules. Unfortunately, despite this relatively high level of involvement we found that the intervention had no apparent effects on either parenting skills or the children’s EBRBs. Using additional strategies to increase engagement in addition to sending two email reminders may have increased the percentage of parents who participated in the intervention, thus increasing the likelihood of revealing an intervention effect. In their supportive accountability model, Mohr et al. argue that human support increases adherence by increasing accountability to a coach who is seen as trustworthy, benevolent, and having expertise, moderated by motivational factors such as the communication media used (e.g., face-to-face, telephone, or email) [84]. However, we expect that parents of children in a high-risk group will be more willing to change their child’s EBRBs, feel a stronger sense of urgency to change their parenting skills, and would therefore be more motivated to use the e-learning program. We would therefore expect a higher percentage of parents to follow the e-learning program, thus revealing intervention effects. Moreover, we expect to find more room for improvement among the high-risk group with respect to improving parenting skills and their child’s EBRBs. On the other hand, if we target a high-risk group who are currently under the care of a health-care professional, it should be easier to increase engagement, as a trusted health-care professional is already involved and can provide the parents with face-to-face support, increasing their likelihood of starting—and ideally, completing—the e-learning program. This healthcare professional can also make follow-up appointments with the parents, and use this appointment also to discuss whether the e-learning program was completed successfully, or —if necessary—encourage them to complete the e-learning program.”.

---

## [Editor Report · Decision Letter 2]

2 Oct 2022

The effectiveness of a web-based Dutch parenting program to prevent overweight in children 9-13 years of age: results of a two-armed cluster randomized controlled trial

PONE-D-21-30893R2

Dear Dr. Ruiter,

Thank you for the work you put into revising this paper. After carefully reading the revisions, your response, and the previous reviews I've decided that this version adequately addresses the concerns and am am pleased to inform you that your manuscript has been judged scientifically suitable for publication and will be formally accepted for publication once it meets all outstanding technical requirements.

Kind regards,

Lee Van Horn, PhD

Academic Editor

PLOS ONE
---

## [Editor Report · Acceptance letter]

13 Oct 2022

PONE-D-21-30893R2 

The effectiveness of a web-based Dutch parenting program to prevent overweight in children 9-13 years of age: results of a two-armed cluster randomized controlled trial 

Dear Dr. Ruiter:

I'm pleased to inform you that your manuscript has been deemed suitable for publication in PLOS ONE. Congratulations! Your manuscript is now with our production department. 

Kind regards, 

on behalf of

Dr. Lee Van Horn 

Academic Editor

PLOS ONE